# Polyoxometalate-based plasmonic electron sponge membrane for nanofluidic osmotic energy conversion

Chengcheng Zhu[1], Li Xu[1], Yazi Liu[2], Jiang Liu [3], Jin Wang[1], Hanjun Sun [1], Ya-Qian Lan [1,3] ✉ & Chen Wang [1] ✉

Nanofluidic membranes have demonstrated great potential in harvesting osmotic energy. However, the output power densities are usually hampered by insufficient membrane permselectivity. Herein, we design a polyoxometalates (POMs)-based nanofluidic plasmonic electron sponge membrane (PESM) for highly efficient osmotic energy conversion. Under light irradiation, hot electrons are generated on Au NPs surface and then transferred and stored in POMs electron sponges, while hot holes are consumed by water. The stored hot electrons in POMs increase the charge density and hydrophilicity of PESM, resulting in significantly improved permselectivity for high-performance osmotic energy conversion. In addition, the unique ionic current rectification (ICR) property of the prepared nanofluidic PESM inhibits ion concentration polarization effectively, which could further improve its permselectivity. Under light with 500-fold NaCl gradient, the maximum output power density of the prepared PESM reaches 70.4 W m$^{-2}$, which is further enhanced even to 102.1 W m$^{-2}$ by changing the ligand to $P_5W_{30}$. This work highlights the crucial roles of plasmonic electron sponge for tailoring the surface charge, modulating ion transport dynamics, and improving the performance of nanofluidic osmotic energy conversion.

The vast osmotic energy derived from the salinity gradient between seawater and river water is an enormously potential blue-energy source, which can be extracted by reverse electrodialysis (RED) that selectively transports ions between electrodes through a permselective membrane[1–3]. As the core component in RED, the permselective membrane plays a fundamental role in nanofluidic osmotic energy harvesting[4,5]. Currently, nanomaterials such as metal organic frameworks (MOFs)[6], covalent organic frameworks (COFs)[7,8], transition metal carbide (MXene)[9], graphene oxide[10], black phosphorus (BP)[11] have been utilized as permselective membranes for RED. The well-controlled mass transport endowed by these membranes have remarkably promoted the advancing of osmotic energy harvesting[4,12]. However, the output power densities are still generally lower than 20 W m$^{-2}$, mainly due to the insufficient permselectivity that increases the loss of Gibbs free energy as Joule heating[4,12–14]. Up to now, several researches have successfully improved the ion selectivity to some extent via tailoring the surface charge like incorporating charged nanomaterials[15,16], grafting functional groups[5,8], or modifying the phase structure[17,18]. However, the current enhancement is far from satisfactory, and the complex preparation process also restricts their applications profoundly. Pursuing efficient strategies to enhance the surface charge of nanofluidic membranes and realize high-performance osmotic conversion are urgently needed.

Polyoxometalates (POMs), also known as electron sponges, can undergo reversible, fast, and stepwise multiple electron-transfer

[1]Jiangsu Key Laboratory of New Power Batteries, School of Chemistry and Materials Science, Nanjing Normal University, Nanjing 210023, China. [2]School of Environment, Jiangsu Engineering Lab of Water and Soil Eco-remediation, Nanjing Normal University, Nanjing 210023, China. [3]School of Chemistry, South China Normal University, Guangzhou 510006, China. ✉e-mail: yqlan@m.scnu.edu.cn; wangchen@njnu.edu.cn

process without changing their structures[19–22]. Dependent on their structures and compositions, POMs have exhibited remarkable properties in diverse applications including energy storage[23], catalysis[19,24], sensing[25] and medicine[26]. Especially, owing to the excellent electron storage and transfer capabilities, POMs can be leveraged to engineer the charge density while maintaining stable structure. However, to the best of our knowledge, POMs have never been applied to develop permselective membranes for osmotic energy harvesting, probably because that most POMs have good water solubility and are difficult to be isolated or recycled from the solutions[27]. In addition, it is also very challenging to target functional materials that could contribute electrons steadily and continuously. Fortunately, POMs can be stably bound to the surface of various metal nanoparticles by electrostatic and steric hindrance effects with the method of ligand exchange strategy[28]. Plasmonic nanoparticles like Au, Ag, Cu, with unique local surface plasmon resonance (LSPR) effect, can provide abundant hot electrons under light irradiation[29,30]. When incorporating plasmonic NPs with POMs, it is highly expected that continuous hot electrons could generate on plasmonic NPs and then transfer to POMs with electrons storage capacity under light irradiation. This process can significantly enhance the negative surface charge with hot holes scavenged by water[31,32], providing a significantly effective strategy to enhance the permselectivity for high-performance osmotic energy conversion.

Herein, we developed a POMs-based plasmonic electron sponge membrane (PESM) as permselective membrane for osmotic energy conversion (Fig. 1). First, a nanofilm of Au@POMs NPs was prepared by virtue of a self-assembly technique at liquid-liquid interface, which were subsequently transferred onto anodic aluminum oxide (AAO) surface, forming Au@POMs/AAO nanofluidic membrane (PESM). Specifically, Au NPs were used as electron donors, $H_3PW_{12}O_{40}$ (denoted as $PW_{12}$ hereafter) as electron acceptors, and AAO is the substrate supporting Au@POMs nanofilm. Under light irradiation, continuous hot electrons generate and transfer from Au NPs to $PW_{12}$ while hot holes are scavenged by water. Accordingly, the as-prepared PESM offers higher charges density and lower energy barrier for ion penetration, leading to enhanced interfacial transport efficiency and boosted osmotic energy conversion. In addition, the inherent structural and charge asymmetries between Au@POMs and AAO render the as-synthesized PESM stable ionic current rectification (ICR) property, which can inhibit ion concentration polarization effectively and further improve ions selectivity and permeability[33–35]. As a result, under light irradiation conditions, the maximum output power density with 500-fold NaCl gradient of the prepared PESMs could reach 70.4 W m$^{-2}$. By using $P_2W_{18}$ and $P_5W_{30}$ ligands, the power density was improved to 79.6 W m$^{-2}$ and 102.1 W m$^{-2}$, respectively. It revealed that by varying the type of polyoxometallates, our approach can be generalized to synthesize ordered hybrid nanostructures with diverse compositions and

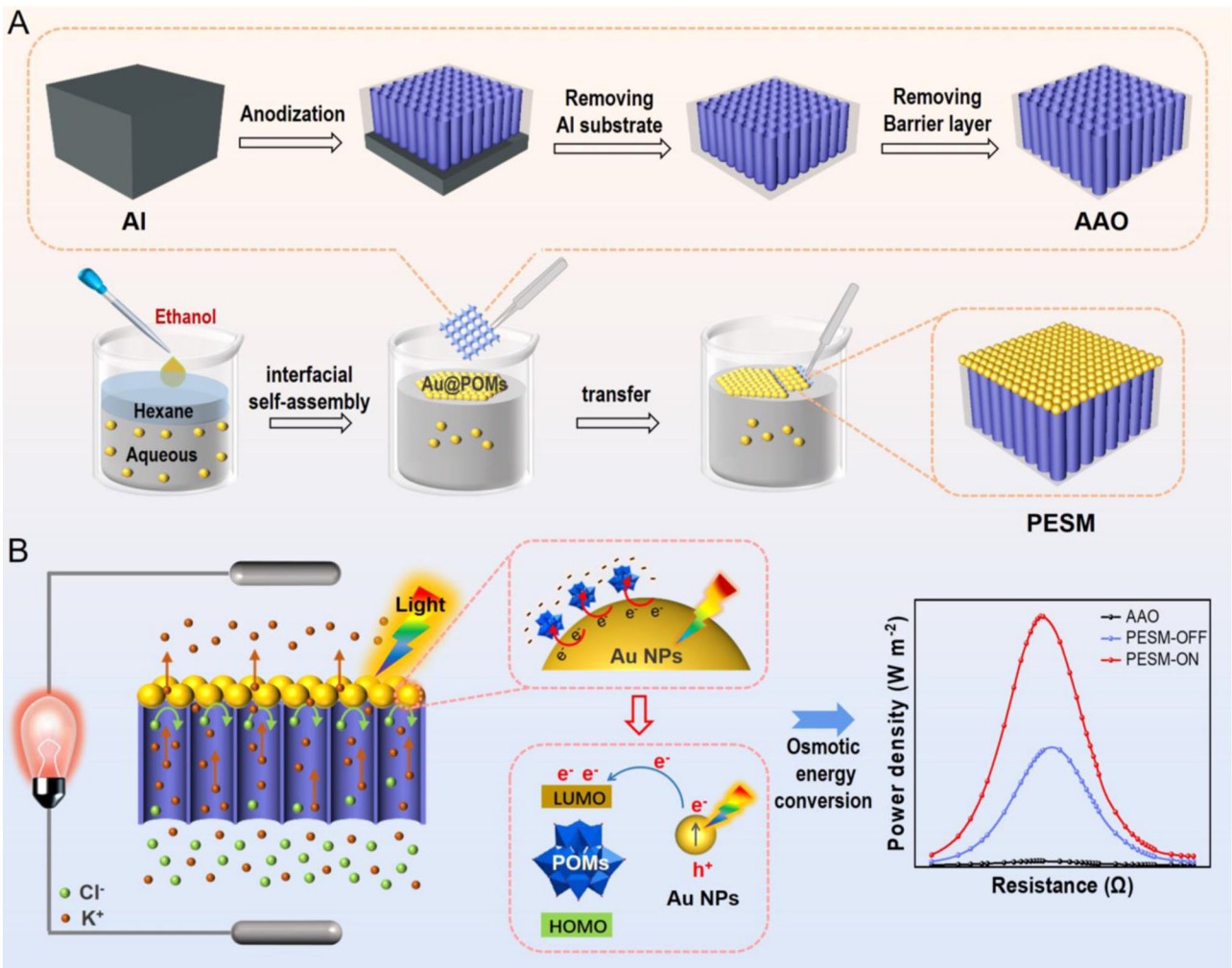

**Fig. 1 | Polyoxometalate-based plasmonic electron sponge membrane (PESM) for nanofluidic osmotic energy conversion. A** Schematic illustration of the fabrication process of anodic aluminum oxide (AAO) and nanofluidic PESM. **B** The application of osmotic energy conversion using the prepared PESM. The orange arrow represents the transport direction of K$^+$. The green arrow represents the transport direction of Cl$^-$.

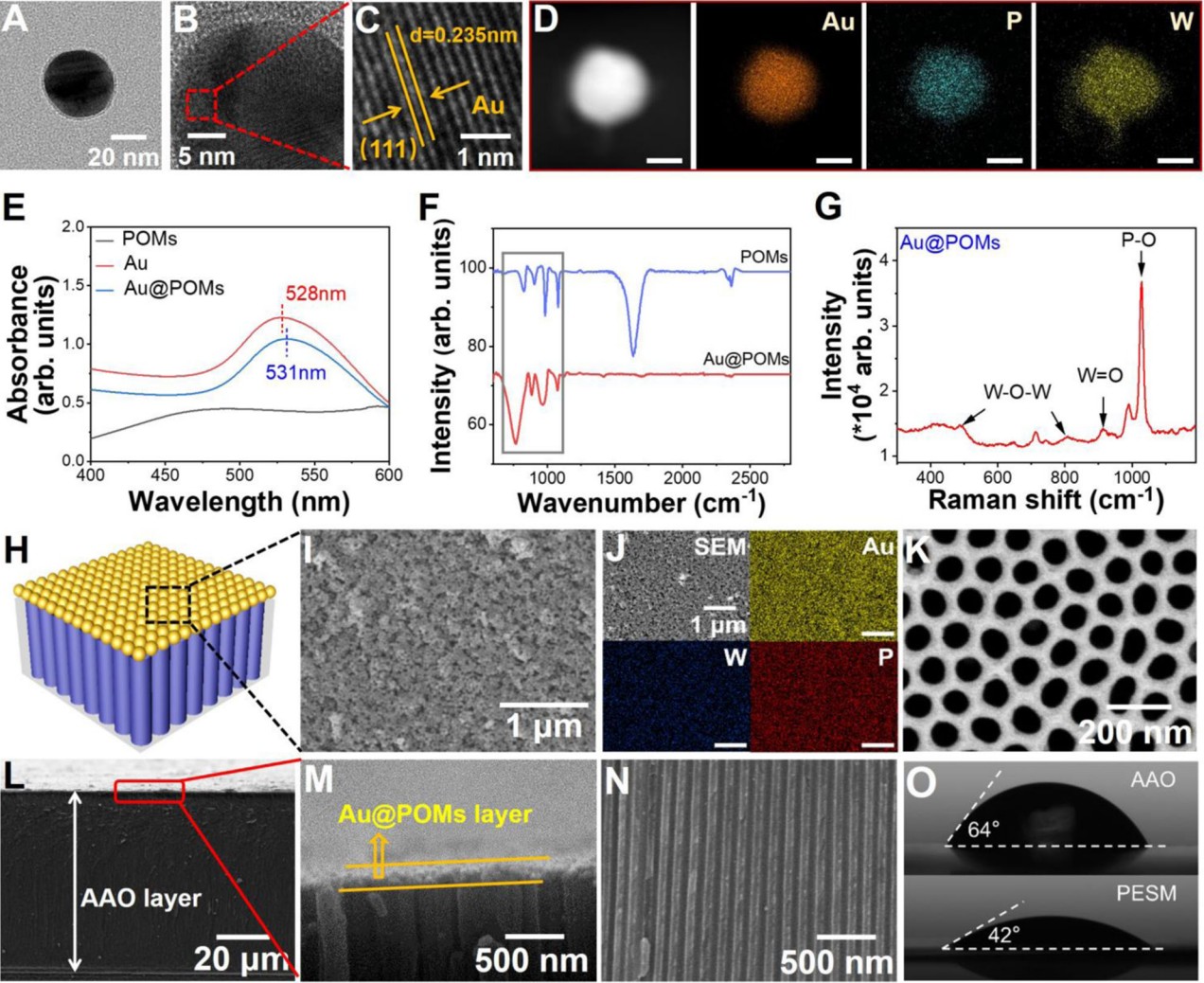

**Fig. 2 | Fabrication and characterization of the PESM. A** The TEM image of Au@POMs. **B, C** HRTEM images of Au@POMs. **D** The HAADF-STEM image of Au@POMs and corresponding energy-dispersive x-ray (EDX) elemental mappings of Au, P and W. (scale bar is 20 nm). **E** UV-Vis absorption spectra of POMs, Au and Au@POMs. **F** FT-IR spectra of POMs and Au@POMs. **G** Raman spectrum of the Au@POMs in aqueous solution. **H** The schematic illustration of PESM. **I** The SEM image of the top of PESM. **J** SEM image of the top of PESM and EDX elemental mappings. **K** The SEM image of the bottom of PESM. **L, M, N** SEM image of the cross-section of PESM. **O** Contact angle of AAO and PESM. Source data are provided as a Source Data file.

morphologies, with far reaching implications for the rational design of nanofluidic membranes.

## Results

### Fabrication and characterization of PESM

PESM was constructed by a simple interfacial self-assembly strategy (Fig. 1) with details illustrated in Experimental Section of SI. First, Au NPs were synthesized via the classical seed-mediated growth method[36]. The transmission electron microscopy (TEM) images showe..d that the diameter of the Au NPs is approximately 35 nm (Supplementary Fig. 1). The X-ray photoelectron spectroscopy (XPS) of pure POMs indicated that the binding energy peaks around 36 eV and 38.2 eV are ascribed to W $4f_{7/2}$ and W $4f_{5/2}$ for W (VI) in its high oxidation state, respectively (Supplementary Fig. 2). Under UV irradiation by isopropyl alcohol, the XPS spectra showed that the binding energies of W $4f_{5/2}$ and W $4f_{7/2}$ in Au@PW$_{12}$ are both negatively shifted by approximately 0.22 eV from the oxidized states to reduced states. Tne results prove the change of the binding energies of W $4f_{5/2}$ and W $4f_{7/2}$ was caused by the reduction of W in the Au@POMs instead of electrostatic binding of POMs and AuNPs[37]. The Au@POMs was then synthesized by mixing reduced POMs with citrate-stabilized Au NPs in

$H_2O$, as demonstrated by TEM. Owing to the excellent contrast of W element in the TEM imaging, a ring-like POMs surrounding the Au NPs could be observed (Fig. 2A and Supplementary Fig. 3). Analysis of the Au@POMs by high-resolution transmission electron microscopy (HRTEM) images indicated that the spacing between two adjacent lattice planes is about 0.235 nm, which agrees well with the spacing of (111) planes of cubic Au (Fig. 2B, C). In addition, the high-angle annular dark-field scanning transmission electron microscopy (HAADF-STEM) image (Fig. 2D) clearly demonstrated that the Au@POMs NPs exhibit a typical core-shell structure. The energy-dispersive x-ray (EDX) elemental mappings further displayed that the element Au is distributed only in the core and the elements P and W of POMs are homogenously distributed throughout the whole NP, revealing the Au NP core is surrounded with a uniform POMs shell.

The Au NPs presented a strong UV-Vis absorption peak centered at 528 nm (Fig. 2E), which is induced by the surface plasmon excitation of Au. Due to electronic interactions between Au and POMs, approximately 3 nm red shift was observed in Au@POMs compared with Au NPs. Fourier-transform infrared (FT-IR) spectroscopy proved grafting of POMs as indicated by the appearance of $v_{sym}(W=O_d)$ and $v_{asym}(W-O_b-W)$ stretching bands (Fig. 2F). Raman spectra displayed

the fingerprint of the POMs at *ca.* 1027.5, 912.2, and 809.4 cm$^{-1}$, which could be assigned to $\nu_{sym}$(P−O), $\nu_{sym}$(W=O$_d$), and $\nu_{asym}$(W−O$_b$−W) vibrations, respectively (Fig. 2G). Furthermore, the zeta-potential measurements indicated that the Au@POMs exhibited more negative value (−41.3 mV) than those of Au NPs (−32.0 mV), indicating higher surface charge density of Au@POMs (Supplementary Fig. 4).

Scanning electron microscopy (SEM) was used to characterize the morphology of AAO membrane (Supplementary Figs. 5 and 6). The fabricated AAO is composed of regular nanochannels with diameter around -75 nm. Large-area uniform monolayer Au@POMs nanofilm was obtained by self-assembly of Au@POMs NPs at a liquid-liquid interface. And the monolayer of Au@POMs film transferred on a silicon wafer substrate was characterized by SEM. As shown in Supplementary Fig. 7, well-ordered Au@POMs nanoparticles are arranged into a relatively dense monolayer structure with little stacking occurring. The obtained Au@POMs was then transferred to the top surface of AAO, forming the PESM (Fig. 2H). The morphology of the as-prepared PESM was characterized by SEM images (Fig. 2I−N). It can be observed that the closely-packed Au@POMs NPs fully covered on top of AAO membrane while no Au@POMs on the bottom surface of AAO. The EDX elemental mappings (Fig. 2J) also showed that Au, P and W elements uniformly distributed on the top part of PESM. SEM images of the cross-section of PESM revealed Au@POMs layer attached onto the top of the AAO membrane (Fig. 2L−N), and the PESM is more hydrophilic than pure AAO (Fig. 2O). In addition, the thickness of PESM could be regulated by repeatedly transferring of Au@POMs layer. Some defects were found in the Au@POMs layer with 70 nm-thickness by transfer once (Supplementary Fig. 8). However, when the Au@POMs layer was transferred twice or more times, no obvious defects could be observed any more. Also, the Current-Voltage (*I-V*) curves display as typical ICR characteristics (Supplementary Figs. 9 and 10), indicating excellent ion selectivity originating from the nanogaps with less than 10 nm among Au@POMs NPs. Considering that larger thickness would lead to weaken ion permeability, the PESM with -115 nm of Au@POMs layer was adopted in the following experiments (Supplementary Figs. 11 and 12). The photograph in Supplementary Fig. 13 indicated that the PESM could be fabricated on a large scale, and the maximum size of PESM depends on the size of the substrate. In this work, the membrane area of PESM is about 6 cm$^2$. Additionally, the prepared film was characterized by SEM at different scales. As shown in Supplementary Fig. 14, SEM images at various magnification fields of view exhibited a homogeneous and flat membrane structure.

## Ion selectivity and permeability of PESM

In order to comprehend the nature of transmembrane ion transport, PESM was sandwiched into a self-made electrochemical cell with two Ag/AgCl electrodes adopted to record the ionic current (Fig. 3A and Supplementary Fig. 15). The typical ionic current rectification (ICR) property occurs for the present PESM, which was caused by ions depletion or accumulation under varied potential bias. In this work, PESM was negatively charged under pH = 7 (Supplementary Fig. 4). When the electric field was applied directing from Au@POMs (top side) to AAO (bottom side), cations (K$^+$) would be driven from the top to bottom while anions (Cl$^-$) flows from bottom to top. Due to the presence of cation-selective Au@POMs layer, it is impossible for Cl$^-$ to pass through the Au@POMs layer freely. As a result, plenty of Cl$^-$ accumulated inside AAO nanochannels. To maintain electroneutrality, the concentration of K$^+$ in the nanochannels would also increase accordingly, leading to higher ion conductance, which was marked as "on" state (Fig. 3B left). In comparison, when the electric field was applied from the bottom to top, the anions (Cl$^-$) were driven from top to bottom, which would be directly excluded by PESM. K$^+$ would be driven from bottom to top, and then flowed through Au@POMs layer smoothly due to the cation-selectivity of PESM. Thus, the concentration of ions within AAO nanochannels was nearly depleted. Under the

above condition, the PESM presented so-called "off" state (Fig. 3B right). Obviously, owing to asymmetry structure and surface charge distribution, PESM displayed better rectification effect as well as larger current responses compared to AAO and Au/AAO (Fig. 3C), rendering excellent ion selectivity and stable ion permeability (Supplementary Figs. 16 and 17)[6,7].

Furthermore, the ICR property was considerably influenced by pH. The ICR ratio ($I_{+1V}/I_{-1V}$) calculated from Supplementary Fig. 14 showed the improved ICR ratio with pH increase (Fig. 3D). To explain the pH-dependent ICR property, the Zeta potential of the PESM were measured as well (Fig. 3D). With increased pH value, the negatively charged Au@POMs nanofilm would gain further negativity, which push more cations through PESM. On the contrary, when pH got lower than 4, the surface charge changed from negative to positive due to the protonation of POMs, leading to a reversed ICR phenomenon (red curve in Supplementary Fig. 18).

To deeply understand the ion transport behaviors of the present PESM, the effect of ion concentration and valence on the ICR effect was systematically investigated (Supplementary Figs. 19−22 and Fig. 3E). Owing to the raised ion strength, the ion current increases with electrolyte concentration. Interestingly, the ICR ratio increased first, then decreased while the electrolyte concentration become larger (Fig. 3E). It has been demonstrated that the thickness of the electric double layer (EDL) is the main factor influencing the magnitude of ICR[38-40]. With the decline of electrolyte concentration, the thickness of EDL increases. The EDL completely overlapped in Au@POMs layer when KCl concentration was less than 0.1 mM, leading to an increased ICR ratio from 0.001 mM (15.03) to 0.1 mM (30.20). In contrast, when KCl concentration was more than 0.1 mM, the incomplete overlap of EDL resulted in the reduction of the ICR ratio from 0.1 mM (30.20) to 100 mM (9.44). When the charge on PESM surface was balanced by the counter-ions in EDL, the ICR ratio reached its maximum value (ICR ratio=30.20). Meanwhile, the ion current also increases with ion valence due to the higher charge density (Supplementary Figs. 20 and 21). The change of ICR ratio is similar to that of KCl, except that the maximum ratio was observed at 0.01 mM (Fig. 3E). Since Ba$^{2+}$ carries charges as twice of the K$^+$, the shielding effect will be stronger. Consequently, the ICR ratio reached the maximum value at a lower electrolyte concentration (0.01 mM, ICR ratio=36.33). The ion transport behavior of MgCl$_2$ was similar to that of BaCl$_2$ owing to the same charges (Supplementary Fig. 23). Similarly, the maximum ICR ratio for K$_2$SO$_4$ is observed at 0.01 mM, with an ICR ratio of 32.7.

Following that, the light-enhanced current of PESM was investigated. The light source used in the experiments is 532 nm from Oxlasers and the light intensity is 200 mW cm$^{-2}$. As revealed by Fig. 3F, the ion current at +1 V dramatically increased from 11.75 μA to 25.61 μA upon light irradiation, which was caused by the photo-induced reduction of the internal resistance of the nanomembrane (Supplementary Fig. 24)[6]. To better evidence this light-induced current change, the current-time curve was recorded under +1 V (red curve) and −1V (black curve) (Fig. 3G). It was clear that the ion current rapidly raised once the light was switched "on" and then reached saturated state within 1 min, indicating the excellent stability and reproducibility of light-switching property of the PESM under different potential. The detailed mechanism and the electron sponge function of PESM will be explained in the following part in detail. Combined with the increased hydrophilicity under irradiation (Supplementary Fig. 25), the conductance of PESM boosted significantly (Fig. 3H), exhibiting the considerably enhanced permeability of PESM under irradiation[41].

## Osmotic energy conversion of PESM

To investigate the performance of osmotic energy conversion, the PESM was placed in an electrochemical cell with the measuring area of -0.03 mm$^2$. Electrolytes of two different concentrations were added into both sides of PESM to confirm the cation selectivity (Fig. 4A). The

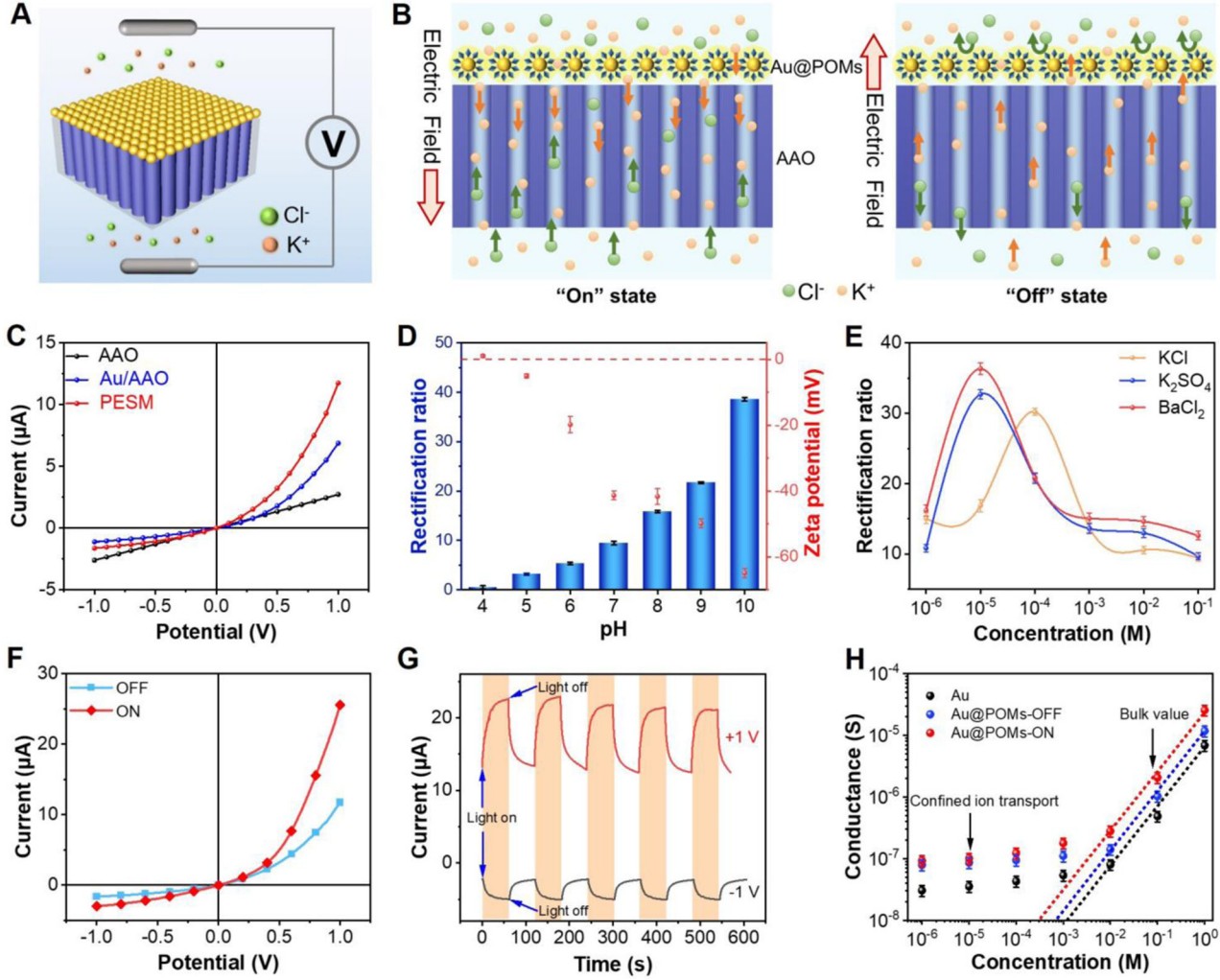

**Fig. 3 | The ion selectivity and permeability of PESM. A** Schematical illustration of PESM for electrochemical measurement. The blue part of the membrane represents AAO and the yellow part of the membrane represents Au@POMs layer. **B** Illustration of the mechanism for ICR. **C** I–V curves of AAO, Au/AAO and PESM in 1 M KCl solution (pH = 7) The orange arrow represents the transport direction of $K^+$. The green arrow represents the transport direction of $Cl^-$. **D** ICR ratio and zeta potential of PESM under different pH values. **E** ICR ratio versus concentration of monovalent electrolyte solution (KCl solution) and divalent electrolyte solutions ($K_2SO_4$ and $BaCl_2$ solution). **F** I-V curves of PESM in 1 M KCl solutions with and without light irradiation. **G** Light-switching property of PESM in 1 M KCl solution. **H** Ion conductance of Au/AAO and PESM with light on and off. The equations of analytical fits are as follows. Au: y = −8.12365x + 0.96674 ($R^2$ = 0.99). Au@POMs-OFF: y = −7.84625x + 0.96121 ($R^2$ = 0.99). Au@POMs-ON: y = −7.57091x + 0.98039 ($R^2$ = 0.99). Error bars represent standard deviation of three different measurements. Source data are provided as a Source Data file.

concentration of KCl electrolyte on the Au@POMs side was 100 mM while AAO side is 0.001 mM. Considering the enormous difference between the concentrations of two solutions, the low-concentration ($C_L$) solution (0.001 mM) at AAO side can hardly contribute to the ion current. Therefore, the ion current is basically contributed by the ion transport from the high-concentration ($C_H$) Au@POMs side to the low-concentration AAO side. When anode was placed at the Au@POMs side, ion current is dominantly contributed by $K^+$ ions. In the same way, $Cl^-$ ions mainly contribute to the ion current when anode was placed at the AAO side. Obviously, PESM possesses cation selectivity because the $K^+$ current was much higher than the $Cl^-$ current (Fig. 4A). Then the performance of osmotic energy conversion was studied by collecting the I-V curves under a series of concentration gradients. The short-circuit current ($I_{SC}$) and open-circuit voltage ($V_{OC}$) can be clearly determined from the intercepts of I-V curves on the current and voltage axis. In fact, the tested $V_{OC}$ is made up of the diffusion potential ($E_{diff}$) and the redox potential ($E_{redox}$). Among them, $E_{diff}$ is generated by the salinity gradient across the nanomembrane, while $E_{redox}$ is generated by the decrease of nonequal potential at the interface

between Ag/AgCl electrodes and solution. $E_{diff}$ can be evaluated by subtracting the $E_{redox}$ from the tested open-circuit potential $V_{OC}$ which followed the Nernst equation (Supplementary Table 1). To seek the optimal direction of ion transport, I-V curves were measured under the gradient of two different salt concentrations (Fig. 4B). When the concentration on AAO side was 1 M ($C_H$) and Au@POMs side was 1 mM ($C_L$) KCl respectively, the internal resistance of the nanomembrane was only 8 KΩ (red line). Lower resistance ensures improved permeability. Thus, high concentration solution was placed on the AAO side in the following experiments. The black and red curves in Fig. 4C evidenced that both $E_{diff}$ and $I_{sc}$ increased with the KCl concentration. The maximum values of $I_{sc}$ and $E_{diff}$ were about 48.4 μA and 123 mV under $C_{high}/C_{low}$ = 3000. The corresponding cationic transfer number of PESM under a series of KCl concentration gradients was recorded in Supplementary Table 2. The cationic transfer number was 0.83 under $C_{high}/C_{low}$ = 1000, which demonstrated PESM carries great cation selectivity in a wide range of concentration.

The output performance of the PESM osmotic energy harvesting system was estimated by connecting it with an external resistor.

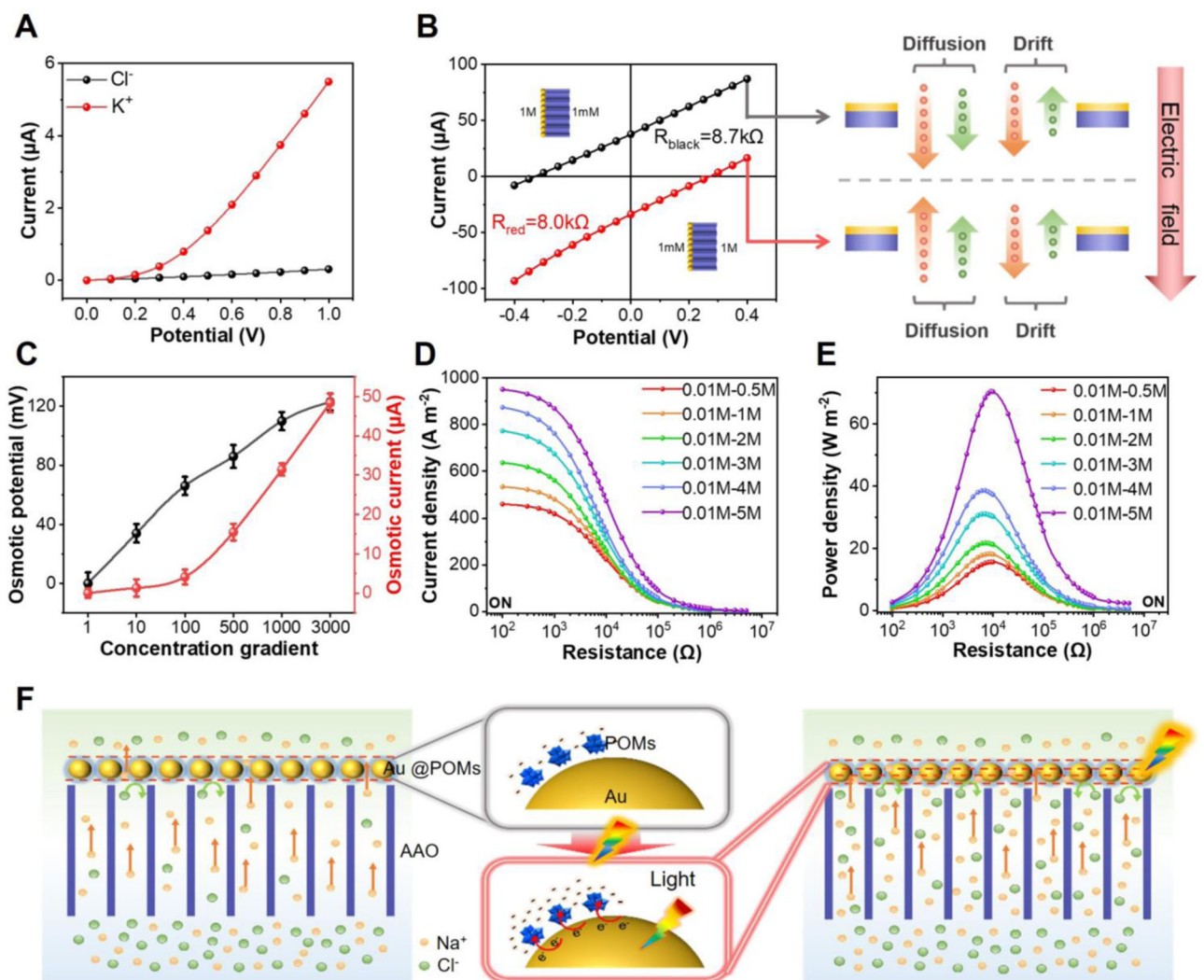

**Fig. 4 | Osmotic energy conversion performance of PESM. A** Ion selectivity of PESM is investigated by measuring the ionic current in two different salt concentrations. 100 mM KCl solution is placed on Au@POMs side and 0.001 mM KCl on AAO side. The black curve represents the current contributed by Cl⁻ and the red curve represents the current contributed by K⁺. **B** I−V curves of the PESM under two different arrangements of the concentration gradient: red curve is achieved by placing high-concentration solution on AAO side, black curve is by placing high-concentration solution on Au@POMs side. **C** Osmotic current and osmotic potential under a series of KCl concentration gradients. Error bars represent standard deviation of three different measurements. **D** Current density of PESM with light irradiation under a series of NaCl concentration gradients. **E** Power density of PESM with light irradiation under a series of NaCl concentration gradients. **F** Illustration of the ion transport mechanism of PESM with and without light irradiation for osmotic energy conversion. The orange arrow represents the transport direction of Na⁺. The green arrow represents the transport direction of Cl⁻. Source data are provided as a Source Data file.

Compared with AAO (1.159 W m⁻²) and Au/AAO (4.89 W m⁻²), the self-assembled PESM could generate the highest power density of 10.27 W m⁻² with external resistance of 20 KΩ under 50-fold NaCl, exceeding the commercial criterion of 5 W m⁻² (Supplementary Figs. 26 and 27). To investigate the effect of concentration gradient on salinity gradient energy conversion, the performance of osmotic energy under different concentrations was evaluated keeping $C_L$ at 0.01 M. Owing to the high driving force, both the current density and the power density increased when concentration gradients raised (Supplementary Fig. 27). The output power density could reach up to 32.55 W m⁻² under 500-fold NaCl, exhibiting a huge potential of the PESM for osmotic energy conversion.

In order to understand the influence of external factors on the harvesting osmotic energy, the output performance of AuNPs size, temperature and pH was investigated. The PESM fabricated by 15 nm AuNPs exhibits high ion transport resistance and weak ionic permeability due to its small nanospace in the Au@POMs layer. In contrast, PESM by 50 nm AuNPs offers a higher ion transport flux while a lower

ion selectivity and rectification. Considering the balance between the selectivity and permeability, AuNPs of 35 nm was chosen, which has the best performance of osmotic energy conversion (Supplementary Fig. 28). The conductance and short-circuit current of PESM increased with the rising temperature due to the low viscosity of the liquid and enhanced ionic mobility, indicating an enhanced ion transport behavior (Supplementary Figs. 29 and 30). On this account, the power density of PESM remained increasing with temperature (Supplementary Fig. 31). However, while the temperature reached ~328 K, the conductance and short-circuit current reduced instead, which might be ascribed to the generated bubbles on the Au@POMs side blocking the ions transport, resulting in reduction of the power density from 22.17 W m⁻² at 328 K to 18.46 W m⁻² at 333 K. Noteworthy, the open-circuit voltage of PESM exhibited a slightly increasing trend with temperature (Supplementary Fig. 32), indicating that ion selectivity would not be weaken when temperature ascended. Meanwhile, pH also has a significant impact on the ion transport of the nanochannel. The conductance of the PESM also increased with pH (Supplementary

Fig. 33) because of the increased surface negative charge density of Au@POMs. As a result, the open-circuit voltage, short-circuit current and output power density increased with pH (Supplementary Figs. 34–36) and the maximal power density could achieve 15.90 W m$^{-2}$ (pH = 10). To sum up, the present PESM can achieve excellent ion transport and energy conversion performance over a wide range of temperature and pH values.

Consideration the importance of irradiation on the osmotic energy conversion, a series of salinity gradient energy conversion of PESM under irradiation was investigated. First, the light intensity has an effect on the energy conversion performance of PESM. The current and power density of PESM increased with the light intensity (Supplementary Figs. 37 and 38). The light intensity was set as 200 mW cm$^{-2}$ in the following experiments. As demonstrated in Supplementary Figs. 39–41, the ion current changes and osmotic energy conversion performance of Au/AAO and PESM were also higher than that of pure AAO membranes. The enhanced performance of Au/AAO could be ascribed to the photothermal effect of Au NPs[42,43], which supported by the IR camera images. As shown in Supplementary Figs. 42 and 43, upon 532 nm laser light irradiation (-200 mW cm$^{-2}$), the temperature of the Au/AAO increased from 25 to 66 °C. Similarly, the temperature of the PESM increased from 26 to 67 °C. It revealed that the photothermal effect of Au/AAO was comparable to that of PESM. While compared with the weak enhancement of Au/AAO, a sharp rise was discerned over PESM for both ion current and output power density under light irradiation. These results indicated that the photothermal effect of PESM contributed only a small part to the enhanced energy conversion performance under irradiation. The experimentally observed light enhancement phenomenon can be ascribed to photoelectric effects. The mechanism of photoelectric effects will be discussed in the next section. The output power density with light irradiation surprisedly achieved a peak value of 15.68 W m$^{-2}$ under 50-fold NaCl, which was 52.7% higher than that without light irradiation (Fig. 4D, E). In addition, the output power density of PESM reached up to 44.25 W m$^{-2}$ at a 400-fold KCl gradient and 70.4 W m$^{-2}$ at a 500-fold NaCl gradient under irradiation (Supplementary Figs. 44–46), indicating the outstanding light-enhanced energy conversion performance of PESM. Under different temperature and pH conditions, the ion current and power density of the PESM were also significantly improved with light on (Supplementary Figs. 47–50). Under light irradiation, continuous hot electrons generate and transfer from Au NPs to PW$_{12}$ while hot holes are scavenged by water. Accordingly, the as-prepared PESM offers higher charges density and lower energy barrier for ion penetration, which benefit the interfacial transport efficiency (Fig. 4F). Therefore, PESM can generate higher transmembrane ionic current by irradiation, resulting in boosted power generation efficiency. To investigate the long-term stability, the PESM membrane was immersed in 10 mM NaCl all the time during 30 days. As displayed in Supplementary Figs. 51 and 52, it still exhibited stable energy harvesting performance after 30 days. Additionally, no visible defects could be observed at the PESM membrane after 30-day usage (Supplementary Figs. 53 and 54), demonstrating high potentail for practical applications. In order to evaluate the practical application value of PESM, the power density under natural seawater (from the sea area near Qingdao) and river water was investigated (Supplementary Fig. 55). Results show that the power density under natural seawater and river water (14.8 W m$^{-2}$) was close to that under 50-fold NaCl (10 mM/500 mM, 15.68 W m$^{-2}$), far exceeding the standard of commercial membrane (5 W m$^{-2}$).

## Mechanism of energy conversion over PESM

To investigate the mechanism of enhanced energy conversion performance under irradiation, the LSPR-induced interaction between Au NPs and POMs was explored. It is well known that a mass of hot carriers could be generated by LSPR excitation[44]. The band structure of the used POMs was estimated by performing cyclic voltammetry measurement[29]. The onset reduction potential appeared at −0.805 V (vs. Ag/AgCl) (Supplementary Fig. 56), which corresponds to the lowest unoccupied molecular orbital (LUMO). The converted LUMO energy level was calculated as −3.905 eV (vs. the vacuum level), more negative than that of hot electrons of Au NPs ($E_{e, hot}$ = −3.49 eV)[45]. Accordingly, hot electrons will tend to transfer from Au NPs to POMs while hot holes will be consumed by water[45], resulting in electron accumulation in the POMs and charge density enhancement (Fig. 5A). The band gap of POMs measured by UV-Vis diffuse reflectance spectrum[29] was displayed to be 3.3 eV (Supplementary Figs. 57 and 58) and the highest occupied molecular orbital (HOMO) of POMs was −7.205 eV (vs. the vacuum level).

To provide direct evidence supporting the above-proposed mechanism, X-ray photoelectron spectroscopy (XPS) experiment was performed. The In-situ XPS spectra indicated that the binding energies of both Au $4f_{5/2}$ and Au $4f_{7/2}$ in Au@POMs were positively shifted by approximately 0.3 eV under irradiation (Fig. 5B). Meanwhile, the binding energies of W $4f_{5/2}$ and W $4f_{7/2}$ in Au@POMs were negatively shifted by approximately 0.2 and 0.3 eV, respectively (Fig. 5C). These results clearly confirmed the electron transfer from Au NPs to POMs under light irradiation. It also indicated the reduction of W in the Au@POMs[37]. In addition, the Au NPs appeared as green spheres in dark field and the LSPR scattering peak of a single particle locates at 588 nm (Fig. 5D). After combined with POMs, the red shift of the scattering peak to 594 nm indicated the decreased electron density on the surface of Au NPs, owing to the injection of hot electrons from Au NPs to POMs. Following that, the open-circuit photovoltage testing was further conducted (Fig. 5E). The potential of the Au@POMs (~ −7 mV) displayed a more negative shift than Au NPs (~ −2 mV) under irradiation, which was attributed to the promoted electron transfer and accumulation between Au NPs and POMs[46]. Furthermore, the Mott-Schottky plots (Supplementary Fig. 59) and Electrochemical impedance spectroscopy (EIS) plots (Fig. 5F) also confirmed electron transfer between Au NPs and POMs, indicating that the plasmon excited "hot electrons" caused a higher charge transport efficiency due to lower resistance of electron transfer[47,48].

To verify the importance of POMs electron sponge in the osmotic energy harvesting performance, additional POMs including P$_2$W$_{18}$ and P$_5$W$_{30}$ with varied electrons storage capacities was tested following the same experiment (Fig. 6A). UV-Vis absorption spectra, XPS spectra and TEM images showed the successful synthesis of Au@POMs using P$_2$W$_{18}$ and P$_5$W$_{30}$ (Supplementary Figs. 60–63). The typical ICR property occurred for all membranes prepared by Keggin-type PW$_{12}$, Wells-Dawson-type P$_2$W$_{18}$ and Preyssler-type P$_5$W$_{30}$ (Fig. 6B). The rsults in Supplementary Figs. 64 and 65 revealed that PESM based on P$_2$W$_{18}$ and P$_5$W$_{30}$ displayed more distinct rectification effect and higher current responses as compared to PW$_{12}$. To further elucidate this phenomenon, we further evaluated the loading capacity and analyzed the molecular orbitals of different POMs with that of AuNPs. First, the level of electrons storage capacity of PW$_{12}$, P$_2$W$_{18}$ and P$_5$W$_{30}$ layer were determined based on the energy-dispersive X-ray spectroscopy analysis of Au@POMs, and the detailed calculation process was provided in Supplementary Table 4[49]. It indicated that the potential charge density of Au@P$_2$W$_{18}$ and Au@P$_5$W$_{30}$ was theoretically higher than Au@PW$_{12}$. Moreover, owing to the decreased energy difference between the energy level of hot electrons of Au NPs ($E_{e, hot}$ = −3.49 eV) and the LUMO level of the POM cluster (Supplementary Figs. 66–70), the as-prepaVis-red PESM based on P$_2$W$_{18}$ and P$_5$W$_{30}$ can offer higher charge density and more asymmetry surface charge distribution. Thus, the enhanced ion transport behaviors over P$_2$W$_{18}$ and P$_5$W$_{30}$ were attributed to the increased electrons storage capacity and charge asymmetry of PESM. Afterwards, the osmotic energy harvesting using varied PESMs was investigated. Results showed that the PESM of P$_5$W$_{30}$ could generate the power density of 30.11 W m$^{-2}$ under 50-fold NaCl

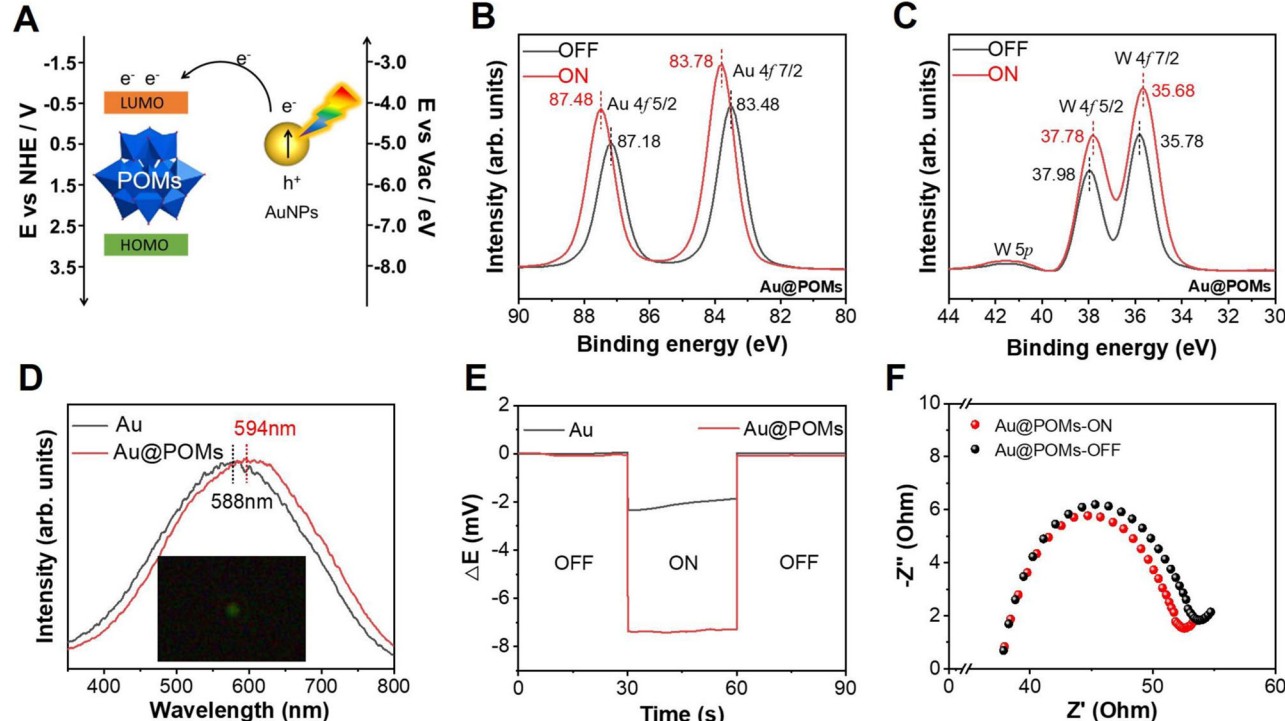

**Fig. 5 | Mechanism of the plasmon-enhanced osmotic energy conversion performance over PESM. A** Schematic and energy level diagram illuminating hot-electron injection from Au NPs to POMs. **B** The Au $4f$ XPS spectra of Au@POMs with and without light irradiation. **C** The W $4f$ XPS spectra of Au@POMs with and without light irradiation. **D** Dark-field scattering spectroscopy of single Au and Au@POMs NP (the intensities are normalized). The inset is dark-field image of a single particle of Au@POMs deposited on an ITO slide. **E** Chronopotentiometry of the $V_{oc}$ from Au and Au@POMs under light irradiation. **F** Nyquist plots of Au@POMs NPs with light on and off collected at open-circuit potential. Source data are provided as a Source Data file.

and 102.1 W m$^{-2}$ under 500-fold NaCl with light irradiation (Fig. 6C, D and Supplementary Figs. 71 and 72), far exceeding that by PW$_{12}$. The output performance of PESM surpassed most of previously reported nanomembranes with analogues operating under similar concentration gradients (50-fold) (Fig. 6E and Supplementary Table 3)[6,9,10,12,13,18,41,50–57]. It is worthy to note that molybdenum (Mo)-based POMs have better performance since they are easier to be reduced (Supplementary Fig. 73). However, considering the cost and easy availability of W-based POMs, we chose PW$_{12}$ as the study model in the present work, which provides a generalized approach to boost the performance of osmotic energy harvesting using POMs and plasmonic materials.

For osmotic energy conversion, fouling remains a significant challenge in the application of membrane technology in real water conditions. The present PESM, which uses PW$_{12}$ as a demonstration, has intrinsic properties such as hydrophilicity, negative charge, rigid smooth surface characteristics and high photo-responsivity. These properties hold great potential for effective antifouling and photo-induced germicidal. As demonstration, Gram-positive bacterium (i.e., *S. aureus*) and Gram-negative bacterium (i.e., *E. coli*) were selected as models and tested by static adhesion tests[58]. The inhibition rates of the PESM reached over 99%, significantly outperforming pure AAO membrane (Fig. 7A and Supplementary Figs. 74 and 75). Importantly, even after biofouling testing, the PESM maintained the power density of ~16 W m$^{-2}$ with a 50-fold NaCl gradient (Fig. 7B). Furthermore, the as-prepared PESM also exhibited an excellent photo-induced germicidal capability approaching ~100% after 300 s of irradiation, which exceeded the performance of commercial polyvinylidene fluoride (PVDF) membranes (Fig. 7C, D and Supplementary Fig. 76). In contrast, the antimicrobial efficiency of PVDF was only 59% and 65% under the same condition, well below the photoinduced germicidal capacity of the present PESM in our work.

## Discussions

In summary, we developed a concept PESM as high-performance osmotic energy conversion device, which is composed of POMs, Au NPs and AAO. Under light irradiation, hot electrons will be generated on Au NPs surface and then be transferred and stored in POMs electron sponges, while hot holes consumed by water. The stored hot electrons in POMs increased the charge density and hydrophilicity of PESM, resulted in significantly improved permselectivity for high-performance osmotic energy conversion. In addition, the unique ICR property of nanofluidic PESM inhibit ion concentration polarization effectively, which could further improve the permselectivity. As demonstration, the as-prepared PESM was applied to osmotic energy harvesting, the maximum output power density of the prepared PESMs could reach 70.4 W m$^{-2}$ with PW$_{12}$, 79.6 W m$^{-2}$ with P$_2$W$_{18}$, and 102.1 W m$^{-2}$ with P$_5$W$_{30}$, respectively. This work uncovers the critical roles of plasmonic electron sponge for tuning the surface charge, regulating ion transport dynamics, promising to be a forerunner in improving the performance of nanofluidic osmotic energy conversion for the alleviation of the energy crisis.

## Methods

### Materials and reagents

Phosphotungstic acid hydrate (PW$_{12}$), Sodium tungstate dihydrate (Na$_2$WO$_4$·2H$_2$O) were purchased from Macklin. The aluminum foil with 99.999% purity and 0.1 mm thickness came from General Research Institute for Nonferrous Metals (Beijing, China). Potassium hydroxide (KOH), hydrochloric acid (HCl), isopropanol, sodium chloride (NaCl), barium chloride (BaCl$_2$), barium chloride (K$_2$SO$_4$), phosphoric acid (H$_3$PO$_4$), Tin (Π) chloride (SnCl$_2$), hydrogen peroxide (30% H$_2$O$_2$), ethanol, Chloroauric acid (HAuCl$_4$) were from Sinopharm Chemical Reagent Co., Ltd. Potassium chloride

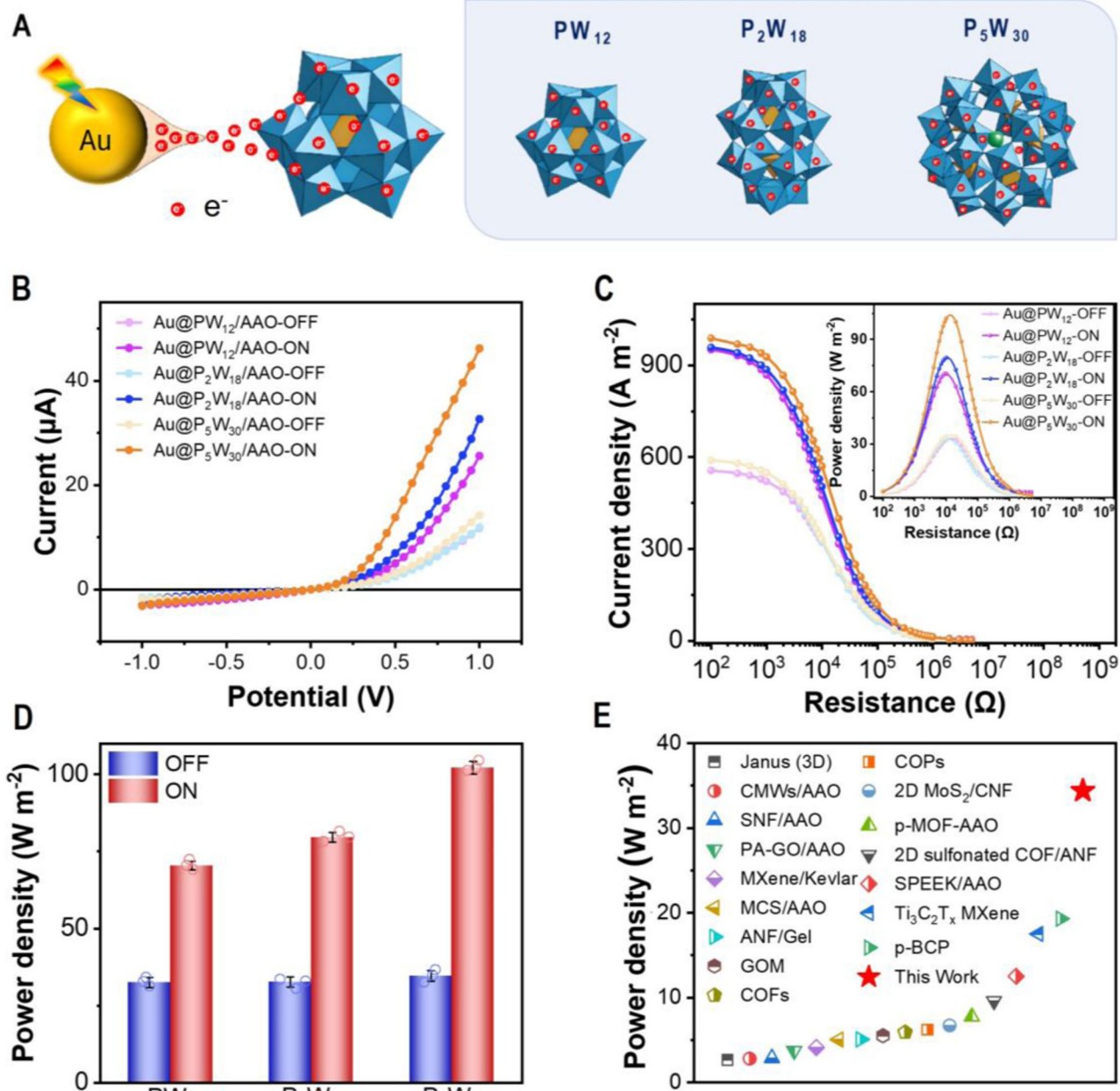

**Fig. 6 | Osmotic energy conversion performance of PESM. A** Schematic illustration of electron transport between from Au NPs to POMs by LSPR excitation, and the crystalline structures of Keggin-type $PW_{12}$, Wells-Dawson-type $P_2W_{18}$ and Preyssler-type $P_5W_{30}$. **B** I–V curves of PESM based on Keggin-type $PW_{12}$, Wells-Dawson-type $P_2W_{18}$ and Preyssler-type $P_5W_{30}$ with and without light irradiation, respectively. **C** Current density of PESM based on Keggin-type $PW_{12}$, Wells-Dawson-type $P_2W_{18}$ and Preyssler-type $P_5W_{30}$ with and without light irradiation, respectively. Concentration gradient is 10 mM/5000 mM NaCl. **D** Power density of PESM based on Keggin-type $PW_{12}$, Wells-Dawson-type $P_2W_{18}$ and Preyssler-type $P_5W_{30}$ with and without light irradiation, respectively. Concentration gradient is 10 mM/5000 mM NaCl. Error bars represent standard deviation of three different measurements. **E** Comparison of the output power density with previously reported devices. Concentration gradient is 50-fold. Source data are provided as a Source Data file.

(KCl), acetone, oxalic acid dehydrate and chromium trioxide ($CrO_3$) were from Shanghai Ling Feng Chemical Reagent Co., Ltd. The resistivity of deionized water was 18.2 MΩ cm⁻¹. The light source used in the experiments is 532 nm from Oxlasers and the light intensity is 50–500 mW cm⁻².

### Instrumentation
The morphology of the top and cross-section of PESM was characterized using a scanning electron microscopy (SEM, S-4800, Hitachi, Japan). The morphology of AuNPs and Au@POM were characterized by transmission electron microscopy (TEM, JEM- 2100, Japan). X-ray diffraction (XRD, SmartLab, Ragaku, Japan) pattern was carried out in the 2θ range of 5° to 90° at room temperature. Zeta potentials were measured by using Malvern Zetasizer Ultra. UV–vis absorption spectra were recorded on an UV-visible spectrophotometer (UV-650 Mapada, Shanghai, China). The water staticcontact angle was measured by a contact angle system (Kruss-DSA25B, Germany). XPS spectra was performed by Kratos AXIS SUPRA (Shimadzu, Japan). The binding energy was calibrated by means of the C 1s peak energy of 284.1 eV. The electrochemical experiments were performed using a CHI660E (Chenhua, China) workstation. Two Ag/AgCl electrodes were used as the anode and cathode.

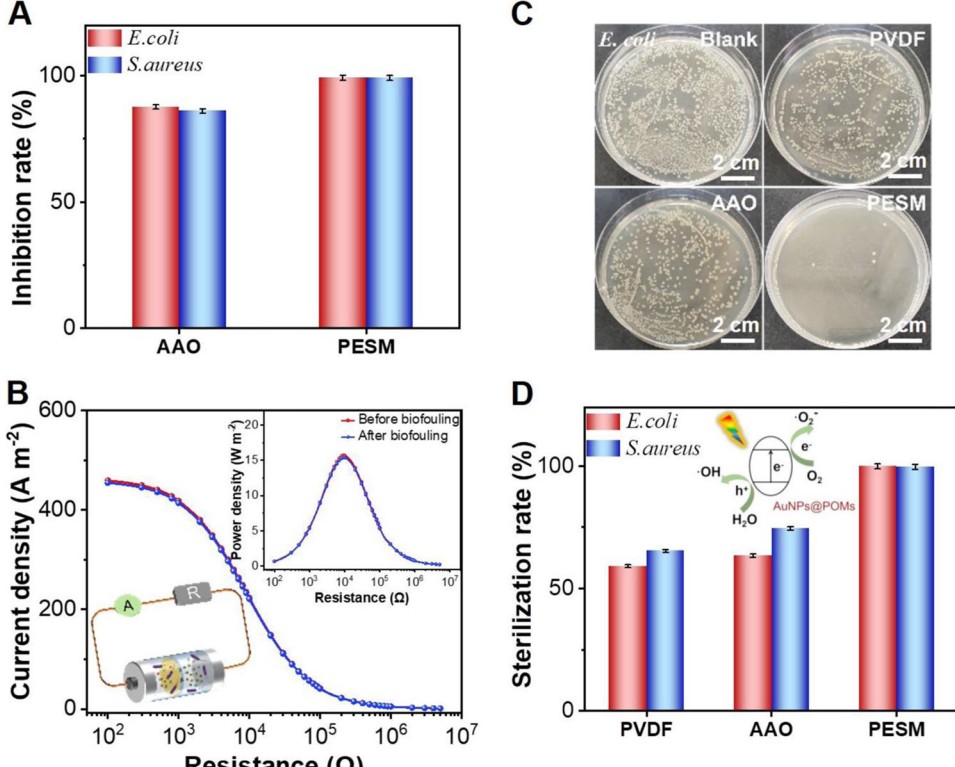

**Fig. 7 | Antifouling and antimicrobial properties of PESM. A** The Inhibition rate of AAO and PESM. **B** Current and power density of PESM before and after biofouling (bacteria attachment) under a 50-fold NaCl gradient. **C** Photographs of bacterial colonies obtained after the photo-induced germicidal test (*E. coli*). **D** The sterilization rate of PVDF, AAO and PESM. Error bars represent standard deviation of three different measurements. Source data are provided as a Source Data file.

## Synthesis of AAO[59]

A two-step anodization method was used to synthesise the AAO membrane[60]. Initially, the aluminum foil (Al) was cleaned with organic solvent and alkali, followed by rinsing with $H_2O$. In the first anodizing step, a voltage of 50 V was applied for half an hour using electrolyte. To remove any irregular oxide layer formed during this step, the first anodized Al was treated with a mixed solution consisting of 6 wt% $H_3PO_4$ and 1.8 wt% $H_2CrO_4$. The second anodization was then carried out for 4 h under the same conditions as the first. The Al substrate was stripped using $SnCl_2$ solution. The AAO membrane was placed in 1.8 wt% $H_3PO_4$ solution for 40 min to remove the barrier layer. The fabricated AAO was then immersed into boiled hydrogen peroxide (30% $H_2O_2$) to generate numerous -OH groups. Finally, it was soaked in $H_2O$ overnight and thoroughly dried.

## Synthesis of 35 nm AuNPs[61]

6 mL of 10 mM $HAuCl_4$ was added to 195 mL of ultrapure water, stirred and then heated at 350 °C. After boiling, 1.5 mL of seed solution (1 nm) and 3.2 mL of 10 mg mL$^{-1}$ trisodium citrate solution were added and thoroughly mixed. Heating and stirring were continued until the color of the solution turned red then heating was discontinued. The solution was stirred at low speed until it reached room temperature.

## Synthesis of PESM[36,62,63]

$PW_{12}$ (2 mL, 2 mM) was added to a beaker, mixed with isopropanol (220 μL), and then irradiated under UV light for 30 min. The concentration of reduced $PW_{12}$ can be controlled by changing the irradiation time. The blue-black solution of reduced $PW_{12}$ and the aqueous solution of 35 nm Au NPs (8 mL) were mixed by shaking gently by hand at room temperature. After several minutes of reaction, PESM was prepared. The prepared Au@POMs solution and n-hexane were added to the beaker and ethanol was then used to self-assemble the Au@POMs nanoparticles into a tightly packed film at the water/n-hexane interface. The PESM was transferred to AAO and heat fixed for 2 h.

## Synthesis of $H_6P_2W_{18}O_{62}\cdot13H_2O$ (P2W18)

50.0 g of $Na_2WO_4\cdot2H_2O$ was dissolved in 60.0 mL of twice distilled water, stirred at room temperature to make it completely dissolved, then 35.0 mL of concentrated $H_3PO_4$ was added, heated and stirred, and the system was milky white for about 20 min. After cooling to room temperature, a certain amount of concentrated HCl was added for acidification. The extraction was carried out with equal volume of ether, and the full shaking was carried out. After standing, the solution was divided into three layers. That's the light yellow $H_6P_2W_{18}O_{62}\cdot13H_2O$.

## Synthesis of $K_{14}[NaP_5W_{30}O_{110}]\cdot22H_2O$ (P5W30)

It was synthesized from a modified procedure[64]. 2 $Na_2WO_4\cdot2H_2O$ (29.7 g) and NaCl (3.51 g) were mixed with deionized water in a 125 mL Teflon-lined autoclave, then 21 mL of 85% $H_3PO_4$ was added as well, the mixture was stirred under room temperature for 1 h. The autoclave was placed in an oven heated at 125 °C for 20 h. After the autoclave was cooled down to room temperature, 9 g of KCl was added to the light-yellow solution and the solution was stirred for 30 min. The produced light-yellow solid was centrifuged under 4400 × g for 5 min. Recrystallization was carried by dissolving this solid in 30 mL deionized water (100 °C, heated in an oil bath). White crystals were formed in next few days (normally within three days) and collected by filtration.

## Electrical measurement

The PESM was securely positioned between the two home-made electrochemical cell halves. Each cell half was then injected with 2 mL of KCl electrolyte. A pair of highly sensitive and stable Ag/AgCl electrodes on CHI660E recorded ionic current. The transmembrane

voltage was adjusted in the range. The effective measurement area for ion transport was -0.1256 mm$^2$. The experiments were carried out at room temperature.

Energy Conversion Measurement: The current density and power density curves were obtained by recording the current under different loaded resistance without external voltage. The generated current was only originated from the osmotic pressure.

Static adhesion tests of bacteria: Adhesion tests of bacteria were conducted to evaluate the antimicrobial performances of PESM, AAO membranes and commercial PVDF membranes. Typical Gram-positive bacteria S. aureus and Gram-negative bacteria *E. coli* were used as model bacteria. The membrane samples with the same diameter of 1 cm were placed into tubes with 10 mL nutrient solution containing a specific concentration of 10$^5$ bacterial suspensions and cultivated with thermostatic oscillation (at $37 \pm 0.5\,°C$, $1000 \times g$) for 24 h. The samples were removed and washed with 20 mL of deionized water.

Then, the sample was put into a beaker containing 10 mL phosphate buffered saline (PBS) for sonication for approximately 3 min. The wash was gathered and continuously diluted to a specific multiple using PBS. 50 μL of the diluted solution was uniformly coated on Luria-Bertani (LB) agar plates and incubated at $37 \pm 0.5\,°C$ for 24 h. The same conditions were used to incubate the bacterial suspension with commercial PVDF membrane samples as a blank control. The colony numbers were counted to calculate the antimicrobial efficiency ($E_a$) as Eq. (1).

$$E_a(\%) = \frac{N_{PVDF} - N_n}{N_{PVDF}} \times 100 \qquad (1)$$

Furthermore, the osmotic energy conversion performance of the composite membrane after bacteria attachment were also measured to evaluate the influence of the antifouling effect.

Photo-induced sterilization tests of bacteria: The photoinduced inactivation endowing positive sterilization performance of composite membranes was also evaluated by the colony-forming units (CFU) method. The composite membranes were coated with 1 mL bacterial suspension ($1 \times 10^7$ cells mL$^{-1}$), followed by exposure to photo irradiation (0.2 W cm$^{-2}$) for 300 s. Bacteria were detached by sonication in 10 mL PBS and the resulting bacterial suspensions were diluted to multiple concentrations. 50 μL of solutions were plated on LB agar plates and incubated at $37 \pm 0.5\,°C$ for 24 h. PVDF membranes under the same conditions and at the same temperature as the composites were used as controls. The same conditions without sterilization were used to incubate the bacterial suspension without membrane samples as a blank control ($N_0$). For each type of surface, the killing efficiency ($E_k$) was calculated as the ratio of the number of colonies (Eq. (2)) in the experimental and blank groups.

$$E_k(\%) = \frac{N_0 - N_n}{N_0} \times 100 \qquad (2)$$

## Data availability

The data that support the conclusions of this study are available from the corresponding authors upon request. Source data are provided with this paper.

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

## Acknowledgements

This work was supported by grants from the National Natural Science Foundation of China (22022412, C. W. and 22274076, C. W.), and the Primary Research & Development Plan of Jiangsu Province (BE2022793, C. W.).

## Author contributions

C.C.Z., Y.Q.L. and C.W. conceived the idea. C.C.Z. conducted most of the experiments. L.X., Y.Z.L., J.L., J.W. and H.J.S. performed some characterization and performance tests. C.C.Z., Y.Z.L., Y.Q.L. and C.W. wrote the paper. All the authors discussed the results and commented on the manuscript.

## Competing interests

Authors declare that they have no competing interests.
