## [Peer Review File · Nature Communications]

POMs-Based Plasmonic Electron Sponge Membrane (PESM) for Nanofluidic Osmotic Energy ConversionREVIEWER COMMENTS

Reviewer #1 (Remarks to the Author):

Nice paper, it reads well, and it is clear.

In my opinion, this paper deserves to be published, although I have a few suggestions.

- 1) I would add the intensity scale in Figure 1E-G;
- 2) I would rethink Figure 2B; the text is clear, but the figure is somehow not(at least to me);
- 3) There are a few typos/formatting errors that need to be addressed.

Reviewer #2 (Remarks to the Author):

Improving the permselectivity of ion-selective membranes is one of the key subjects for achieving high power density and the practical application of salinity-gradient power generation systems. This manuscript presented a strategy of designing ion-selective Au@POMs/AAO membranes with high permselectivity by exploiting the electron storage property of POM to store the electrons generated from the LSPR effect of Au nanoparticles. Therefore, osmotic power density was much promoted because of the increased charge density of the membrane under light irradiation. In general, the strategy was fairly verified by the experiments. The following major points need to be properly addressed ahead of consideration for publication.

1. What is the light source and the light intensity used in the experiments? It's not mentioned throughout the manuscript. How is the influence of light intensity on the current and the power density? It is recommended that the authors clarify this by experimental verifications, to better evaluate the actual performance of the membrane and its practical application potential for salinity-gradient power generation.
2. Besides LSPR effect, the Au nanoparticles also possess photo-thermal conversion capability. Photothermal effect has been widely proved to enhance ion transport and thereby improve power density. Will the photothermal effect also contribute to the enhanced power generation here in this case? The authors are recommended to testify this by taking comparing experiments. For example, the photothermal property of the PESM membrane and the Au/AAO membrane, the ion current of Au/AAO membrane with and without light irradiation.
3. As most POMs are quite water soluble as mentioned in the manuscript, are the POMs used in this work water stable? What is the combining force between POMs and Au nanoparticles? How are the POMs combined onto Au NPs? These questions have not been discussed in detail in the text, however are important for readers to understand the design principle of the material. If it has been reported before, proper discussion and reference citation are also necessary.
4. Why is Ba²⁺ rather than the commonly used Mg²⁺ or Ca²⁺ used during the study of ion valence on the ICR effect?
5. How were the long-term power generation experiments carried out (Figure S42 and S43)? Was the membrane immersed in electrolyte solutions all the time during 30 days? The authors need to provide detailed description on the experimental process. In addition, long-term photo-induced current are better characterized to confirm the stability of the photo-promoted ion transport and power generation. As mechanical and water stability of the membrane is key for the long-term usage and the practical application, the authors are also encouraged to provide large-scale SEM characterizations of the membrane before and after 30-day usage.
6. In the first paragraph of the Introduction part, hydrogels are not nanomaterials.
7. In figure 2G, there are two curves, which difference and meaning was not specifically labeled and discussed in the caption or in the text.

Reviewer #3 (Remarks to the Author):

Aiming at the issue of nanofluidic membranes pursuing efficient strategies to enhance the surface

charge and realize high-performance osmotic conversion, the authors proposed a route to use polyoxometalates (POMs)-based nanofluidic plasmonic electron sponge membrane (PESM) for highly efficient osmotic energy conversion. Combined with experimental characterizations and computational simulations, the authors revealed that POMs can serve as electron sponges to transfer and store hot electrons of Au NPs under light conditions, to significantly improve the permselectivity of osmotic energy conversion with high performance. Under 500 times NaCl gradient, the maximum output power densities of PW12, P2W18, and P5W30 achieve excellent results of 70.4 W/m², 79.6 W/m², and 102.1 W/m², respectively. In contrast, the overall analysis of the results seemed to be not rigorous enough. The understanding of the working mechanism should be further addressed and demonstrated. As a result, I suggest that the manuscript go through some necessary revisions. Specific comments are as follows:

1. In page 4 line 101, it was described "The Au@POMs was then synthesized by mixing reduced POMs with citrate stabilized Au NPs in H₂O, as demonstrated by TEM." The authors chose reduced POMs as the electron sponge but did not characterize the reduced state accordingly. On the other hand, the W4f binding energy in XPS spectra of Au@POMs in Figure 4C should give an analysis of the reduced state. The reduced states of the other two POMs, P2W18 and P5W30 mentioned in the manuscript, were also not stated correspondingly. The significance and function of the reduced state should have been discussed correspondingly.

2. In page 4 line 127, it was mentioned, "Large-area uniform monolayer Au@POMs nanofilm was obtained by self-assembly of Au@POMs NPs at a liquid-liquid interface." A suitable structural characterization for the monolayers should have been carried out.

3. The Au@POMs NPs nanomembranes were prepared by virtue of a self-assembly technique at the liquid-liquid interface. The method is simple. However, the maximum size and uniformity of the prepared film affect the practical application of the material, which should be supplemented accordingly.

4. In page 15 line 367, it was stated "Owing to asymmetry structure and surface charge distribution, PESH based on P2W18 and P5W30 displayed more distinct rectification effect and higher current responses as compared to PW12." This description was not explained suitably. The performance of P5W30 seemed better than that of PW12. The evaluation of the loading capacity of different POMs and the analysis of the molecular orbitals of different POMs with that of Au NPs should be addressed.

5. The article tried to evaluate the power density under the condition of a single salt solution. In order to demonstrate the practical application value of Au@POMs membrane, the power density versus the salinity difference between natural seawater and river water seems to be considered as well.

6. In Figure 3F, the author gives an Illustration of the ion transport mechanism of PESH with and without light irradiation. The concentration of salt solution on both sides of the figure is inconsistent with the concentration selected in the experiment. The reverse case should be more appropriate.

7. It is suggested to double-check the format of the whole article. (a) "As demonstration, the as-prepared PESH was applied to osmotic energy harvesting, and the maximum output power density of the prepared PESHs could reach 70.4 W/m² with PW12, 79.6 W/m² with P2W18, 102.1 W/m² with P5W30, respectively. This work highlights the crucial roles of plasmonic electron sponge for tailoring the surface charge, modulating ion transport dynamics, and improving the performance of nanofluidic osmotic energy conversion." This passage is repeated in the Abstract, Introduction, and Conclusion parts. The highlights, significance, and application prospects can be further elaborated; (b) Figure 1E is in the wrong reference location. It should be on page 5 line 117, not page 7, line 161; and (c) Figure 2G lacks identification of two curves.

RESPONSE TO REVIEWERS' COMMENTS

Reviewer #1 (Remarks to the Author):

In my opinion, this paper deserves to be published, although I have a few suggestions.

1) I would add the intensity scale in Figure 1E-G;

Response: Thanks for the reviewer's suggestion. The revised figures are shown in **Figure R1** below.

Figure R1. A) UV-vis absorption spectra of POMs, Au and Au@POMs. B) FT-IR spectra of POMs and Au@POMs. C) Raman spectrum of the Au@POMs in aqueous solution.

We have added the intensity scales of Figure 1E-G in the revised version. All the above revisions are highlighted in yellow.

2) I would rethink Figure 2B; the text is clear, but the figure is somehow not (at least to me);

Response: Thanks for the reviewer's suggestion. The revised figure is shown in **Figure R2** below.

Figure 2. Illustration of the mechanism for ICR.

We have modified Figure 2B to make it clearer in the revised version. The revised version was highlighted in yellow.

3) There are a few typos/formatting errors that need to be addressed.

Response: Thanks for the reviewer's kind reminder. We have carefully checked the whole paper and corrected the typos/formatting errors.

Reviewer #2 (Remarks to the Author):

Improving the permselectivity of ion-selective membranes is one of the key subjects for achieving high power density and the practical application of salinity-gradient power generation systems. This manuscript presented a strategy of designing ion-selective Au@POMs/AAO membranes with high permselectivity by exploiting the electron storage property of POM to store the electrons generated from the LSPR effect of Au nanoparticles. Therefore, osmotic power density was much promoted because of the increased charge density of the membrane under light irradiation. In general, the strategy was fairly verified by the experiments. The following major points need to be properly addressed ahead of consideration for publication.

1. What is the light source and the light intensity used in the experiments? It's not mentioned throughout the manuscript. How is the influence of light intensity on the current and the power density? It is recommended that the authors clarify this by experimental verifications, to better evaluate the actual performance of the membrane and its practical application potential for salinity-gradient power generation.

Response: Thanks for the reviewer's valuable comments. The light source used in the experiments is 532 nm from OxLasers and the light intensity is 200 mW/cm². We have supplemented the relevant experiment to evaluate the influence of light intensity on the current and the power density. As shown in **Figure R3**, the results indicated that the current and power density increased with the increasing light intensity.

Figure R3. The (A) current density and (B) power density of PESN under different light intensity.

We have added the above results in the revised manuscript on page 10 lines 232-234 and page 13 lines 323-326, and highlighted them in yellow. The above figures have been added as **Figure S37** (page S26) in supporting information.

Page 10 line 232-234: *The light source used in the experiments is 532 nm from Oxlasers and the light intensity is 200 mW/cm².*

Page 13 lines 323-326: *First, the light intensity has an effect on the energy conversion performance of PESM. The current and power density of PESM increased with the light intensity (Figures S37-S38). The light intensity was set as 200 mW/cm² in the following experiments.*

2. Besides LSPR effect, the Au nanoparticles also possess photo-thermal conversion capability. Photothermal effect has been widely proved to enhance ion transport and thereby improve power density. Will the photothermal effect also contribute to the enhanced power generation here in this case? The authors are recommended to testify this by taking comparing experiments. For example, the photothermal property of the PESM membrane and the Au/AAO membrane, the ion current of Au/AAO membrane with and without light irradiation.

Response: Thanks for the reviewer's valuable comments. According to the reviewers' suggestions, we carefully considered and explored the influence of the photothermal effect on enhanced energy conversion. First, we compared the photothermal properties of the PESM membrane and the Au/AAO membrane. As shown in **Figures R4-R5**, upon 532 nm laser light irradiation (~200 mW/cm²), the temperature of the Au/AAO increased from 25 to 66 °C. Similarly, the temperature of the PESM increased from 26 to 67 °C. It revealed that the photothermal effect of Au/AAO is comparable to that of PESM. Besides, the ion current and power density of Au/AAO and PESM were measured with and without light irradiation, respectively. As displayed in **Figure R6**, the ion current of Au/AAO was also enhanced to some degree after light irradiation, while the enhancement was far less significant than that of PESM. And the enhanced energy conversion performance under irradiation is shown in **Figures R7-R8**, demonstrating that the power density changes in Au/AAO before and after light irradiation are not evident. These results revealed that the photothermal effect of Au NPs could enhance ion transport and osmotic energy conversion, but the effect was not significant. While compared with Au/AAO, a sharp rise was discerned over PESM for both ion current and output power density under light irradiation, which indicated that the photothermal effect of PESM contributes only a small part to the enhanced energy conversion performance under irradiation.

Figure R4. IR camera images (532 nm, $\sim 200 \text{ mW/cm}^2$) of Au/AAO.

Figure R5. IR camera images (532 nm, $\sim 200 \text{ mW/cm}^2$) of PESM.

Figure R6. Current changes of AAO, Au/AAO and PESM with and without light irradiation, respectively.

Figure R7. (A) Current density and (B) power density of AAO, Au/AAO and PESM with and without light irradiation, respectively. Concentration gradient is 10 mM/500 mM NaCl.

Figure R8. Power density of AAO, Au/AAO and PESM with and without light irradiation, respectively. Concentration gradient is 10 mM/500 mM NaCl.

We have added **Figures R4** and **R5** as **Figures S42** and **S43** in the revised supporting information (Pages S27-S29). The detailed explanations have been added in the revised manuscript on pages 13-14 lines 326-33. All the above revisions are highlighted in yellow.

Pages 13-14 lines 326-339: *As demonstrated in **Figures S39-41**, the ion current changes and osmotic energy conversion performance of Au/AAO and PESM were also higher than that of pure AAO membranes. The enhanced performance of Au/AAO could be ascribed to the photothermal effect of Au NPs,^{42,43} which supported by the IR camera images. As shown in **Figures S42 and S43**, upon 532 nm laser light irradiation (~ 200 mW/cm²), the temperature of the Au/AAO increased from 25 to 66 °C. Similarly, the temperature of the PESM increased from 26 to 67 °C. It revealed that the photothermal effect of Au/AAO was comparable to that of PESM. While compared with the weak enhancement of Au/AAO, a sharp rise was discerned over PESM for both ion current and output power density under light irradiation. These results indicated that the photothermal effect of PESM contributed only a small part to the enhanced energy conversion performance under irradiation. The experimentally observed light enhancement phenomenon can be ascribed to photoelectric effects. The mechanism of photoelectric effects will be discussed in the next section.*

3. As most POMs are quite water soluble as mentioned in the manuscript, are the POMs used in this work water stable? What is the combining force between POMs and Au nanoparticles? How are the POMs combined onto Au NPs? These questions have not been discussed in detail in the text, however are important for readers to understand the design principle of the material. If it has been reported before, proper discussion and reference citation are also necessary.

Response: Thanks for the reviewer's valuable comments. As reported, most POMs are quite water soluble, including the POMs used in this work. However, many effective strategies have been developed to improve the water stability of POMs, including support (J. Am. Chem. Soc. 2009, 131, 17412-17422), encapsulation (Angew. Chem. Int. Ed. 2020, 59, 20779-20793), covalent attachment (J. Am. Chem. Soc. 2022, 144,

1861-1871). In this study, the method of support was utilized. Specifically, using citrate-protected 35 nm-diameter gold nanoparticles as the support substrate, we prepared POM layers by ligand exchange on Au NPs. Ultimately, the POMs were stably bound to the surface of Au NPs by electrostatic and steric hindrance effects. (J. Am. Chem. Soc. 2003, 125, 8440-8441).

We have added the detailed explanations in the revised version on page 3 lines 64-65. The relevant reference was cited as ref 28. All the above revisions are highlighted in yellow.

Page 3 lines 64-65: Fortunately, POMs can be stably bound to the surface of various metal nanoparticles by electrostatic and steric hindrance effects with the method of ligand exchange strategy.²⁸

Ref 28: Wang, Y., Neyman, A., Arkhangelsky, E., Gitis, V., Meshi, L., Weinstock, I. A. Self-Assembly and Structure of Directly Imaged Inorganic-Anion Monolayers on a Gold Nanoparticle. *J. Am. Chem. Soc.* **131**, 17412-17422 (2009).

4. Why is Ba^{2+} rather than the commonly used Mg^{2+} or Ca^{2+} used during the study of ion valence on the ICR effect?

Response: Thanks for the reviewer's comment. The ions Ba^{2+} , Mg^{2+} and Ca^{2+} are all often utilized when investigating the influence of ionic valence on ICR effect. Here the Ba^{2+} was selected as an example. Following the reviewer's suggestion, we have conducted additional experiments using various concentrations of MgCl_2 . As shown in **Figure R9**, the ionic currents of MgCl_2 and BaCl_2 at the same concentration are similar. The maximum ICR ratio of MgCl_2 reaches at the same concentration as BaCl_2 , consistent with predicted results.

Figure R9. The (A) I-V curves and (B) ICR ratio of PESN in various concentrations of MgCl_2 solution.

We have added **Figure R9** as **Figure S23** in the revised supporting information. The corresponding descriptions have been added in the manuscript on page 9 lines 220-221, and highlighted in yellow.

Page 9 lines 220-221: *The ion transport behavior of MgCl₂ was similar to that of BaCl₂ owing to the same charges (Figure S23).*

5. How were the long-term power generation experiments carried out (Figure S42 and S43)? Was the membrane immersed in electrolyte solutions all the time during 30 days? The authors need to provide detailed description on the experimental process. In addition, long-term photo-induced current are better characterized to confirm the stability of the photo-promoted ion transport and power generation. As mechanical and water stability of the membrane is key for the long-term usage and the practical application, the authors are also encouraged to provide large-scale SEM characterizations of the membrane before and after 30-day usage.

Response: Thanks for the reviewer's suggestions. In the long-term power generation experiments, the PESM membrane was constantly immersed in 10 mM NaCl all the time for 30 days. The long-term photo-induced current was measured, and it was found that it still had steady output power density after soaking in 10 mM NaCl for 30 days (Figure R10). Moreover, to confirm the mechanical and water stability of the membrane, The large-scale SEM characterizations of the membrane before and after 30-day usage were performed. As shown in Figures R11-R12, no obvious defects could be observed at the PESM membrane after 30-day usage, which validated the feasibility of long-term usage and the practical application.

Figure R10. Stability of PESM under irradiation for energy conversion (50-fold NaCl). The PESM membrane was constantly immersed in 10 mM NaCl all the time for 1-30 days, respectively.

Figure R11. The photograph of the PESM membrane after 30-day usage. The PESM membrane was constantly immersed in 10 mM NaCl all the time for 30 days.

Figure R12. (A-D) The SEM images of the PESM membrane after 30-day usage. The PESM membrane was constantly immersed in 10 mM NaCl all the time for 30 days.

We have added **Figures R11** and **R12** as **Figures S53** and **S54** in the revised supporting information (page S34). The detailed explanations were added in the revised manuscript on page 14 lines 352-357 and highlighted in yellow.

Page 14 lines 352-357: *To investigate the long-term stability, the PESM membrane was immersed in 10 mM NaCl all the time during 30 days. As displayed in **Figures S51 and S52**, it still exhibited stable energy harvesting performance after 30 days. Additionally, no visible defects could be observed at the PESM membrane after 30-day usage (**Figures S53 and S54**), demonstrating high potential for practical applications.*

6. In the first paragraph of the Introduction part, hydrogels are not nanomaterials.

Response: Thanks for the reviewer's kind reminder. We have corrected this mistake in the revised manuscript. The sentence was revised as "Currently, nanomaterials such as metal organic frameworks (MOFs),⁶ covalent organic frameworks (COFs),^{7,8} transition metal carbide (MXene),⁹ graphene oxide,¹⁰ black phosphorus (BP)¹¹ have been utilized as permselective membranes for RED."

7. In figure 2G, there are two curves, which difference and meaning was not specifically labeled and discussed in the caption or in the text.

Response: Thanks for the reviewer's kind reminder. We have modified **Figure 2G** in the revised manuscript, also shown as **Figure R13 below**. Specifically, the current-time curve was recorded under +1 V (red curve) and -1V (black curve) to better evidence this light-induced current change. It was clear that the ion current rapidly raised once the light was switched "on" and then reached a saturated state within 1 min, indicating the excellent stability and reproducibility of the light-switching property of the PESM

under different potential.

Figure R13. Light-switching property of PESM in 1 M KCl solution.

The supplementary descriptions were also added in the revised manuscript on page 10 lines 236-238 and highlighted in yellow.

Page 10 lines 236-238: *To better evidence this light-induced current change, the current-time curve was recorded under +1 V (red curve) and -1V (black curve) (Figure 2G).*

Reviewer #3 (Remarks to the Author):

Aiming at the issue of nanofluidic membranes pursuing efficient strategies to enhance the surface charge and realize high-performance osmotic conversion, the authors proposed a route to use polyoxometalates (POMs)-based nanofluidic plasmonic electron sponge membrane (PESM) for highly efficient osmotic energy conversion. Combined with experimental characterizations and computational simulations, the authors revealed that POMs can serve as electron sponges to transfer and store hot electrons of Au NPs under light conditions, to significantly improve the permselectivity of osmotic energy conversion with high performance. Under 500 times NaCl gradient, the maximum output power densities of PW₁₂, P₂W₁₈, and P₅W₃₀ achieve excellent results of 70.4 W/m², 79.6 W/m², and 102.1 W/m², respectively. In contrast, the overall analysis of the results seemed to be not rigorous enough. The understanding of the working mechanism should be further addressed and demonstrated. As a result, I suggest that the manuscript go through some necessary revisions. Specific comments are as follows:

1. In page 4 line 101, it was described “The Au@POMs was then synthesized by mixing reduced POMs with citrate stabilized Au NPs in H₂O, as demonstrated by TEM.” The authors chose reduced POMs as the electron sponge but did not characterize the reduced state accordingly. On the other hand, the W4f binding energy in XPS spectra of Au@POMs in Figure 4C should give an analysis of the reduced state. The reduced states of the other two POMs, P₂W₁₈ and P₅W₃₀ mentioned in the manuscript, were also not stated correspondingly. The significance and function of the reduced state should have been discussed correspondingly.

Response: Thanks for the reviewer's valuable comments. As recommended, the XPS spectra of POMs (including PW_{12} , P_2W_{18} and P_5W_{30}) and Au@POMs have been added in the revised manuscript and supporting information. As displayed in **Figures R14-R16**, the binding energy peaks around 36-36.2 eV and 38.2-38.3 eV are ascribed to W 4f_{7/2} and W 4f_{5/2} for W (VI), respectively. It revealed that the state of W in POMs (including PW_{12} , P_2W_{18} , and P_5W_{30}) is in its high oxidation state. Notably, the XPS spectra of Au@ PW_{12} showed that the binding energies of W 4f_{5/2} and W 4f_{7/2} are both negatively shifted by approximately 0.22 eV, as compared to those in PW_{12} (**Figures R14 and R17**). Similarly, the binding energies of W 4f_{5/2} and W 4f_{7/2} in Au@ P_2W_{18} are negatively shifted by approximately 0.4 eV as compared to those in P_2W_{18} (**Figure R15**). And the binding energies of W 4f_{5/2} and W 4f_{7/2} in Au@ P_5W_{30} are negatively shifted by approximately 0.4 eV as compared to those in P_5W_{30} (**Figure R16**). These results could certify the reduction of W in the Au@POMs according to the literature (Nat. Commun., 2019, 10:1330). Notably, the reduced state is of great significance because exposure of photochemically reduced POMs to Au NPs resulted in the formation of stable metal nanoparticles capped by POMs.

Figure R14. The W 4f XPS spectra of PW_{12} and Au@ PW_{12} .

Figure R15. The W 4f XPS spectra of P_2W_{18} and Au@ P_2W_{18} .

Figure R16. The W 4f XPS spectra of P_5W_{30} and $Au@P_5W_{30}$.

Figure R17. The W 4f XPS spectra of $Au@POMs$ with and without light irradiation.

We have added **Figures R14 -R16** as **Figures S2, S62 and S63** in the revised supporting information (Pages S38-S39). And the detailed explanations were added in the manuscript on page 5 lines 105-110 and page 16 lines 407-409. All the above revisions are highlighted in yellow.

Page 5 lines 105-110: *The X-ray photoelectron spectroscopy (XPS) of pure POMs indicated that the binding energy peaks around 36 eV and 38.2 eV are ascribed to $W 4f_{7/2}$ and $W 4f_{5/2}$ for $W(VI)$ in its high oxidation state, respectively (**Figure S2**). Under UV irradiation by isopropyl alcohol, the XPS spectra showed that the binding energies of $W 4f_{5/2}$ and $W 4f_{7/2}$ in $Au@PW_{12}$ are both negatively shifted by approximately 0.22 eV, certifying the reduction of W in the $Au@POMs$.³⁷*

Page 16 lines 407-409: *UV-vis absorption spectra, XPS spectra and TEM images showed the successful synthesis of $Au@POMs$ using P_2W_{18} and P_5W_{30} (**Figures S60-S63**).*

2. In page 4 line 127, it was mentioned, “Large-area uniform monolayer $Au@POMs$ nanofilm was obtained by self-assembly of $Au@POMs$ NPs at a liquid-liquid interface.” A suitable structural characterization for the monolayers should have been carried out.

Response: Thanks for the reviewer’s kind suggestion. We have characterized the monolayer of $Au@POMs$ film transferred on a silicon wafer substrate by SEM. As shown in **Figure R18**, well-ordered $Au@POMs$ nanoparticles are arranged into a relatively dense monolayer structure with little stacking occurring.

Figure R18. Large-area uniform monolayer Au@POMs nanofilm transferred once on the Silicon wafer.

We have added **Figure R18** as **Figure S7** in the revised supporting information (Page S11). The detailed descriptions were supplemented in the manuscript on page 6 lines 138-141. All the above revisions are highlighted in yellow.

Page 6 lines 138-141: *And the monolayer of Au@POMs film transferred on a silicon wafer substrate was characterized by SEM. As shown in **Figure S7**, well-ordered Au@POMs nanoparticles are arranged into a relatively dense monolayer structure with little stacking occurring.*

3. The Au@POMs NPs nanomembranes were prepared by virtue of a self-assembly technique at the liquid-liquid interface. The method is simple. However, the maximum size and uniformity of the prepared film affect the practical application of the material, which should be supplemented accordingly.

Response: Thanks for the reviewer's valuable comment. According to the reviewer's suggestion, we supplemented the maximum size of the prepared films and characterized their homogeneity by SEM. The photograph in **Figure R19** indicates that the PESM can be fabricated on a large scale, and the maximum size of PESM depends on the size of the substrate. In this work, the membrane area of PESM is about 6 cm². Additionally, we characterized the prepared film by SEM at different scales. As shown in **Figure R20**, SEM images at various magnification fields of view exhibited a homogeneous and flat membrane structure.

Figure R19. The photograph of the PESM membrane.

Figure R20. (A-D) The SEM images of the PESM membrane.

We have added **Figures R19** and **R20** as **Figures S13** and **S14** in the revised supporting information (Page S11). The detailed explanations were added in the manuscript on page 6 lines 157-162. All the above revisions are highlighted in yellow.

Page 6 lines 157-162: *The photograph in **Figure S13** indicated that the PESM could be fabricated on a large scale, and the maximum size of PESM depends on the size of the substrate. In this work, the membrane area of PESM is about 6 cm². Additionally, the prepared film was characterized by SEM at different scales. As shown in **Figure S14**, SEM images at various magnification fields of view exhibited a homogeneous and flat membrane structure.*

4. In page 15 line 367, it was stated “Owing to asymmetry structure and surface charge distribution, PESM based on P₂W₁₈ and P₅W₃₀ displayed more distinct rectification effect and higher current responses as compared to PW₁₂.” This description was not explained suitably. The performance of P₅W₃₀ seemed better than that of PW₁₂. The evaluation of the loading capacity of different POMs and the analysis of the molecular orbitals of different POMs with that of Au NPs should be addressed.

Response: Thanks for the reviewer’s valuable comment. Accordingly, we have evaluated the loading capacity and analyzed the molecular orbitals of different POMs with that of Au NPs. First, the TEM and energy-dispersive X-ray spectroscopy (EDS) analysis of Au@POMs (with different ligands: PW₁₂, P₂W₁₈ and P₅W₃₀) was performed, respectively (**Table R1**). The loading capacity of different POMs was evaluated by the mass fraction of Au and W (Angew. Chem. Int. Ed. 2022, e202205873), and the detailed calculation process was provided in the supporting information as below: “*The Number of POMs Loaded around AuNPs is calculated using the formula:*

$$N = \frac{\rho(\text{Au}) \times V(\text{Au}) \times N_A \times \omega(\text{W})}{M(\text{W}) \times \omega(\text{Au}) \times N(\text{W})}$$

V(Au) represent the volume of one AuNPs. M(W) represent the atomic mass of W. N(W)

represent the number of *W* atom in one POMs.

Based on the energy-dispersive X-ray spectroscopy analysis of Au@POMs (with different ligands: PW_{12} , P_2W_{18} and P_5W_{30}), the loading capacity of POMs was evaluated by the mass fraction of Au and W, and the detailed calculation process was provided in Table S4.⁴⁹ The average number of PW_{12} per AuNP was calculated to be approximately 7341. Similarly, the average number of P_2W_{18} and P_5W_{30} per AuNP was calculated to be approximately 5563 and 3550, respectively. Correspondingly, the level of electrons storage capacity of PW_{12} , P_2W_{18} and P_5W_{30} layer were determined to be 88092, 100134, and 106500 e^- , respectively. It indicated that the potential charge density of Au@ P_2W_{18} and Au@ P_5W_{30} was theoretically higher than Au@ PW_{12} .”

Moreover, owing to the decreased energy difference between the energy level of hot electrons of Au NPs ($E_{e, hot} = -3.49$ eV) and the LUMO level of the POM cluster ($E_{LUMO, P_2W_{18}} = -3.903$ eV, $E_{LUMO, P_5W_{30}} = -3.891$ eV) (**Figure R21-R25**), the as-prepared PESM based on P_2W_{18} and P_5W_{30} offers higher charges density and more asymmetry surface charge distribution. It resulted in a more distinct rectification effect and higher current responses as compared to PW_{12} . Collectively, the performance of PESM based on P_5W_{30} was better than that of PW_{12} .

Figure R21. The cyclic voltammogram of the POMs (P_2W_{18}) using an Ag/AgCl reference electrode.

Figure R22. The band gap of POMs (P_2W_{18}).

Figure R23. The cyclic voltammogram of the POMs (P_5W_{30}) using an Ag/AgCl reference electrode.

Figure R24. The band gap of POMs (P_5W_{30}).

Figure R25. Schematic and energy level diagram illustrating hot-electron injection from Au NPs to POMs (including PW_{12} , P_2W_{18} and P_5W_{30}).

	PW_{12}			P_2W_{18}			P_5W_{30}		
Mass Fraction of Au (%)	93.9	94.24	94.34	93.25	93.51	93.48	93.04	93.07	92.96
Mass Fraction of W (%)	6.1	5.76	5.66	6.75	6.49	6.52	6.96	6.93	7.04
The Number of POMs Loaded around AuNPs	7688	7234	7101	5711	5476	5503	3541	3525	3585
The Average Number of POMs	7341			5563			3550		
The Number of Electrons Received per Cluster	12 e^-			18 e^-			30 e^-		
The Maximum number of electrons per Au@POMs	88092 e^-			100134 e^-			106500 e^-		

Table R1. TEM transmission electron microscopy (TEM) combined with energy-dispersive X-ray spectroscopy (EDS) analysis of Au@POMs using PW_{12} , P_2W_{18} and P_5W_{30} as ligands, respectively²³.

Remarkably, our previous expressions were not very accurate, so the sentence mentioned has been revised on page 16. And the **Figures R21-R25** were added in the revised supporting information as **Figures S66-S70** (Pages S40-S42), and the **Table R1** was as **Table S4** (Page S47). We also added the relevant descriptions in the manuscript

on pages 16-17 lines 411-424. All the above revisions are highlighted in yellow.

Pages 16-17 lines 411-424: *The results in Figures S64 and S65 revealed that PESM based on P_2W_{18} and P_5W_{30} displayed more distinct rectification effect and higher current responses as compared to PW_{12} . To further elucidate this phenomenon, we further evaluated the loading capacity and analyzed the molecular orbitals of different POMs with that of AuNPs. First, the level of electrons storage capacity of PW_{12} , P_2W_{18} and P_5W_{30} layer were determined based on the energy-dispersive X-ray spectroscopy analysis of Au@POMs, and the detailed calculation process was provided in Table S4.⁴⁹ It indicated that the potential charge density of Au@ P_2W_{18} and Au@ P_5W_{30} was theoretically higher than Au@ PW_{12} . Moreover, owing to the decreased energy difference between the energy level of hot electrons of Au NPs ($E_{e, hot} = -3.49$ eV) and the LUMO level of the POM cluster (Figures S66-S70), the as-prepared PESM based on P_2W_{18} and P_5W_{30} can offer higher charge density and more asymmetry surface charge distribution. Thus, the enhanced ion transport behaviors over P_2W_{18} and P_5W_{30} were attributed to the increased electrons storage capacity and charge asymmetry of PESM.*

5. The article tried to evaluate the power density under the condition of a single salt solution. In order to demonstrate the practical application value of Au@POMs membrane, the power density versus the salinity difference between natural seawater and river water seems to be considered as well.

Response: Thanks for the reviewer's suggestion. The power density under natural seawater (from the sea area near Qingdao) and river water was added in the revised version. As shown in Figure R26, the power density under natural seawater and river water was close to that under 50-fold NaCl (10 mM/500 mM), which just as reported that the salinity gradient between natural seawater and river water is about 50-fold (Nat. Commun., 2023, 14:5926).

Figure R26. The current density and power density of the PESM membrane under natural seawater and river water.

We have added Figure R26 as Figure S55 (Page S35) in the revised supporting information. And the detailed explanations were added in the revised manuscript on page 14 lines 357-362. All the above revisions are highlighted in yellow.

Page 14 lines 357-362: *In order to evaluate the practical application value of PESM, the power density under natural seawater (from the sea area near Qingdao) and river*

water was investigated (**Figure S55**). Results show that the power density under natural seawater and river water (14.8 W/m^2) was close to that under 50-fold NaCl ($10 \text{ mM}/500 \text{ mM}$, 15.68 W/m^2), far exceeding the standard of commercial membrane (5 W/m^2).

6. In Figure 3F, the author gives an illustration of the ion transport mechanism of PESM with and without light irradiation. The concentration of salt solution on both sides of the figure is inconsistent with the concentration selected in the experiment. The reverse case should be more appropriate.

Response: Thanks for the reviewer's kind reminder. We have corrected this mistake in Figure 3F in the revised manuscript, also shown in **Figure R27** below.

Figure R27. Illustration of the ion transport mechanism of PESM with and without light irradiation.

7. It is suggested to double-check the format of the whole article. (a) “As demonstration, the as-prepared PESM was applied to osmotic energy harvesting, and the maximum output power density of the prepared PESMs could reach 70.4 W/m^2 with PW_{12} , 79.6 W/m^2 with P_2W_{18} , 102.1 W/m^2 with P_5W_{30} , respectively. This work highlights the crucial roles of plasmonic electron sponge for tailoring the surface charge, modulating ion transport dynamics, and improving the performance of nanofluidic osmotic energy conversion.” This passage is repeated in the Abstract, Introduction, and Conclusion parts. The highlights, significance, and application prospects can be further elaborated; (b) Figure 1E is in the wrong reference location. It should be on page 5 line 117, not page 7, line 161; and (c) Figure 2G lacks identification of two curves.

Response: Thanks for the reviewer's kind reminder.

(a) We have reorganized the repeated passages in the Abstract, Introduction, and Conclusion parts. And the highlights, significance, and application prospects were further elaborated. All the above revisions are highlighted in yellow. The details are as follows:

Abstract: Under light with 500-fold NaCl gradient, the maximum output power density of the prepared PESM reaches 70.4 W/m^2 , which is further enhanced even to 102.1 W/m^2 by changing the ligand to P_5W_{30} . This work highlights the crucial roles of plasmonic electron sponge for tailoring the surface charge, modulating ion transport dynamics, and improving the performance of nanofluidic osmotic energy conversion.

Introduction: As a result, under light irradiation conditions, the maximum output power density with 500-fold NaCl gradient of the prepared PESMs could reach 70.4 W/m^2 . By using P_2W_{18} and P_5W_{30} ligands, the power density was improved to 79.6 W/m^2 and 102.1 W/m^2 , respectively. It revealed that by varying the type of polyoxometallates, our approach can be generalized to synthesize ordered hybrid nanostructures with diverse compositions and morphologies, with far reaching implications for the rational design of nanofluidic membranes.

Conclusion: As demonstration, the as-prepared PESM was applied to osmotic energy harvesting, the maximum output power density of the prepared PESMs could reach 70.4 W/m^2 with PW_{12} , 79.6 W/m^2 with P_2W_{18} , and 102.1 W/m^2 with P_5W_{30} , respectively. This work uncovers the critical roles of plasmonic electron sponge for tuning the surface charge, regulating ion transport dynamics, promising to be a forerunner in improving the performance of nanofluidic osmotic energy conversion for the alleviation of the energy crisis.

(b) We have corrected the reference location of **Figure 1E** on page 5 line 199 in the revised manuscript.

(c) We have added the identification of two curves in **Figure 2G**, also shown in **Figure R28** below. The current-time curve was recorded under +1 V (red curve) and -1V (black curve) to better evidence this light-induced current change.

Figure R28. Light-switching property of PESM in 1 M KCl solution.

REVIEWER COMMENTS

Reviewer #2 (Remarks to the Author):

The authors have revised the manuscript carefully, and have provided detailed responses to my comments. This manuscript could be accepted for publication in Nat Commun now.

Reviewer #3 (Remarks to the Author):

The authors made a series of reasonable explanations to previous questions and improved the manuscript with the supplemented experiments. A few small cares seemed to need explanation in a bit more detail so a wider readership can better understand the advantages of this work.

1.The authors supplemented the XPS spectra of POMs (including PW12, P2W18, and P5W30) and Au@POMs, and explained the reduced states of POMs by citing the literature. However, some small problems still need to be explained. The authors chose to compare the binding energies of W 4f_{5/2} and W 4f_{7/2} in oxidized POMs and reduced Au@POMs, but no obvious peaks belonging to W(V) are observed, which could not rule out the change caused by electrostatic binding of POMs and AuNPs. It is suggested to add the XPS spectra of reduced POMs and oxidized Au@POMs.

2.Why chose this size AuNPs and the effect of the size on the nanofluidic osmotic energy conversion?

3.The tungsten-based POMs were chosen as electron sponges. Considering the easier reduction of molybdenum(Mo)-based POMs, will they have better performance?

RESPONSE TO REVIEWERS' COMMENTS

Reviewer #3 (Remarks to the Author):

The authors made a series of reasonable explanations to previous questions and improved the manuscript with the supplemented experiments. A few small cares seemed to need explanation in a bit more detail so a wider readership can better understand the advantages of this work.

1. The authors supplemented the XPS spectra of POMs (including PW_{12} , P_2W_{18} , and P_5W_{30}) and Au@POMs, and explained the reduced states of POMs by citing the literature. However, some small problems still need to be explained. The authors chose to compare the binding energies of W 4f_{5/2} and W 4f_{7/2} in oxidized POMs and reduced Au@POMs, but no obvious peaks belonging to W(V) are observed, which could not rule out the change caused by electrostatic binding of POMs and AuNPs. It is suggested to add the XPS spectra of reduced POMs and oxidized Au@POMs.

Response: Thanks for the reviewer's valuable comments. We have supplemented the XPS spectra to evaluate the binding energies of W 4f_{5/2} and W 4f_{7/2} in reduced POMs and oxidized Au@POMs. The revised figures are shown in **Figure R1** below.

The binding energy of the W 4f peak shifted negatively from the oxidized states to reduced states (Figure R1). Taking PW_{12} as an example, the W4f peak in Au@ PW_{12} is consistent with the reduced state PW_{12} . When Au@ PW_{12} is oxidized, the location of the W4f peak is restored to an identical level with that of PW_{12} . According to the previous study (*J. Am. Chem. Soc.* 2009, 131, 17412-17422), the POM-anion shells were obtained by ligand exchange from citrate-protected gold nanoparticles and the POM monolayer growth occurs via "islands". Above all, it revealed that the change of the binding energies of W 4f_{5/2} and W 4f_{7/2} was caused by the reduction of W in the Au@POMs instead of electrostatic binding of POMs and AuNPs. The similar methods and explanations have been reported by previous studies (*J. Am. Chem. Soc.* 2009, 131, 17412-17422; *Nanoscale Adv.*, 2019, 1,3400).

We have added the XPS spectra of reduced POMs and oxidized Au@POMs in the

revised **Figures S2&S62&S63**. The revised version was highlighted in yellow.

Figure R1. (A) The XPS spectra of PW_{12} and $Au@PW_{12}$. (B) The XPS spectra of P_2W_{18} and $Au@P_2W_{18}$. (C) The XPS spectra of P_5W_{30} and $Au@P_5W_{30}$.

2. Why chose this size AuNPs and the effect of the size on the nanofluidic osmotic energy conversion?

Response: Thanks for the reviewer's question. To choose a proper size of AuNPs, some additional experiments were added in the revised version.

First, AuNPs of 15 nm and 50 nm were synthesized, and the corresponding characterizations was shown below (**Figures R2A and B**). The $Au@POMs$ using 15 nm and 50 nm AuNPs were then synthesized by mixing reduced POMs with citrate-stabilized AuNPs in H_2O , as demonstrated by TEM (**Figures R2C and D**). The energy-dispersive x-ray (EDX) elemental mappings further revealed that the element Au was distributed only in the core and the elements P and W of POMs were homogeneously distributed throughout the whole NP (**Figures R2E and F**). The PESM using 15 nm and 50 nm AuNPs were constructed by a simple interfacial self-assembly strategy. As shown in **Figures R2G and H**, no obvious defects could be observed in either of the two newly synthesized PESMs.

Then, we evaluated the effect of AuNPs size of on the performance of nanofluidic osmotic energy conversion and ion current rectification. The I-V curves indicated that with the particle size increasing, the current increased while current rectification decreased (**Figures R3A and B**). The results were consistent with the previous studies

(*Angew. Chem. Int. Ed.* 2022, 61, e202202698; *Nat. Rev. Mater.* 2021, 6, 622-639), which is caused by the effect of size and surface charge density. Specifically, The PESM fabricated by 15 nm AuNPs exhibits high ion transport resistance and weak ionic permeability due to its small nanospace in the Au@POMs layer. In contrast, PESM by 50 nm AuNPs offers a higher ion transport flux while a lower ion selectivity and rectification. Considering the balance between the selectivity and permeability, AuNPs of 35 nm was chosen, which has the best performance of osmotic energy conversion (**Figure R3C**).

We have added the above results in the revised manuscript on **page 12 line 285**, and highlighted them in yellow. The above figures have been added as **Figure S28** in supporting information.

Figure R2. (A) TEM image of 15 nm AuNPs. (B) TEM image of 50 nm AuNPs. (C) TEM image of Au@POMs using 15 nm AuNPs. (D) TEM image of Au@POMs using 50 nm AuNPs. (E) The HAADF-STEM image of Au@POMs using 15 nm AuNPs and corresponding energy-dispersive x-ray (EDX) elemental mappings of Au, P and W. (F) The HAADF-STEM image of Au@POMs using 50 nm AuNPs and corresponding energy-dispersive x-ray (EDX) elemental mappings of Au, P and W. (G) SEM image and EDX elemental mappings of the top of PESM based on 15 nm AuNPs. (H) SEM image and EDX elemental mappings of the top of PESM based on 50 nm AuNPs.

Figure R3. (A) I-V curves of PESM based on 15 nm, 35 nm and 50 nm AuNPs. (B) ICR ratio of PESM based on 15 nm, 35 nm and 50 nm AuNPs. (C) The current and power density of PESM based on 15 nm, 35 nm and 50 nm AuNPs (10 mM/500 mM NaCl).

3. The tungsten-based POMs were chosen as electron sponges. Considering the easier reduction of molybdenum (Mo)-based POMs, will they have better performance?

Response: Thanks for the reviewer's valuable comments. In the revised version, we supplemented the relevant experiment to evaluate the performance of Mo-based PESM. First, the Au@PMo₁₂NPs was synthesized using the 35 nm-AuNPs by ligand exchange strategy, and characterized by TEM images and the EDX elemental mapping (**Figures R4A-C**). As shown in **Figures R4D and E**, the Au@PMo₁₂ layer was covered on AAO surface, forming the Mo-based PESM. The Au, P and Mo elements uniformly distributed on the top part of PESM (**Figure R4F**). To investigate the performance of Mo-based PESM, the I-V curves and output power density were evaluated under the same condition with W-based PESM (**Figures R4G and H**). The results showed that PMo₁₂-based PESM had higher interfacial transport efficiency and boosted osmotic energy conversion than PW₁₂, owing to the easier reduction of Mo-based POMs. Due to the easy accessibility and low cost of W-based POMs, we used W-POMs in our work. The present study reveals that the method we proposed can be extended to prepare other tungsten-based POMs, providing a generalized approach to boost the performance of osmotic energy harvesting systems using POMs and plasmonic materials.

We have added the above results in the revised manuscript on **page 17 lines 425-427**, and highlighted them in yellow. The above figures have been added as **Figure S73** in supporting information.

Figure R4. (A&B) TEM images of Au@PMo₁₂. (C) The HAADF-STEM image of Au@PMo₁₂ and corresponding energy-dispersive x-ray (EDX) elemental mappings of Au, P and Mo. (D) SEM image of the top of PESM based on PMo₁₂. (E) SEM image of the cross-section of PESM based on PMo₁₂. (F) SEM image and EDX elemental mappings of the top of PESM based on PMo₁₂. (G) I-V curves of PESM based on PMo₁₂ in 1M KCl solutions with and without light irradiation. (H) The current and power density of PESM based on PMo₁₂ with and without light irradiation (10 mM/500 mM NaCl).

REVIEWERS' COMMENTS

Reviewer #3 (Remarks to the Author):

The authors made a very good explanation. The revisions are acceptable and therefore, the article is publishable in its present status.